# High reactivity of deep biota under anthropogenic CO$_2$ injection into basalt

Rosalia Trias[1,2], Bénédicte Ménez [1,2], Paul le Campion[1,2], Yvan Zivanovic[3], Léna Lecourt[1,2], Aurélien Lecoeuvre[1], Philippe Schmitt-Kopplin[4], Jenny Uhl[4], Sigurður R. Gislason[5], Helgi A. Alfreðsson[5], Kiflom G. Mesfin[5], Sandra Ó. Snæbjörnsdóttir[5], Edda S. Aradóttir[6], Ingvi Gunnarsson[6], Juerg M. Matter[7,8], Martin Stute[8,9], Eric H. Oelkers[5,10,11] & Emmanuelle Gérard[1,2]

Basalts are recognized as one of the major habitats on Earth, harboring diverse and active microbial populations. Inconsistently, this living component is rarely considered in engineering operations carried out in these environments. This includes carbon capture and storage (CCS) technologies that seek to offset anthropogenic CO$_2$ emissions into the atmosphere by burying this greenhouse gas in the subsurface. Here, we show that deep ecosystems respond quickly to field operations associated with CO$_2$ injections based on a microbiological survey of a basaltic CCS site. Acidic CO$_2$-charged groundwater results in a marked decrease (by ~ 2.5–4) in microbial richness despite observable blooms of lithoautotrophic iron-oxidizing Betaproteobacteria and degraders of aromatic compounds, which hence impact the aquifer redox state and the carbon fate. Host-basalt dissolution releases nutrients and energy sources, which sustain the growth of autotrophic and heterotrophic species whose activities may have consequences on mineral storage.

[1] Institut de Physique du Globe de Paris, Sorbonne Paris Cité, Univ. Paris Diderot, CNRS UMR 7154, 1 rue Jussieu, 75238 Paris cedex 05, France. [2] Centre de Recherches sur le Stockage Géologique du CO2 (IPGP/TOTAL/SCHLUMBERGER/ADEME), 1 rue Jussieu, 75238 Paris cedex 05, France. [3] Institut de Biologie Intégrative de la Cellule UMR 9198, Univ. Paris-Sud, 15 rue Georges Clémenceau, 91405 Orsay cedex, France. [4] Helmholtz-Zentrum Muenchen-German Research Center for Environmental Health, Research Unit Analytical Biogeochemistry, Ingolstaedter Landstrasse 1, 85764 Neuherberg, Germany. [5] Institute of Earth Sciences, University of Iceland, Sturlugötu 7, 101 Reykjavik, Iceland. [6] Reykjavik Energy, Baejarhals 1, 110 Reykjavik, Iceland. [7] Ocean and Earth Science, University of Southampton, University Road, Southampton, SO17 1BJ, UK. [8] Lamont-Doherty Earth Observatory, Columbia University, 61 Route 9w, Palisades, NY 10964, USA. [9] Barnard College, 3009 Broadway, New York, NY 10027, USA. [10] GET-UMR 5563, Univ.Paul-Sabatier, IRD, 14 avenue Edouard Belin, 31400 Toulouse, France. [11] Earth Sciences, University College London, Gower Street, London, WC1E 6BT, UK. Correspondence and requests for materials should be addressed to B.M. (email: menez@ipgp.fr) or to E.G. (email: emgerard@ipgp.fr)

CO₂ geological storage at depth in depleted oil reservoirs, saline aquifers or (ultra)mafic rocks is considered as a solution, which may reconcile the use of fossil fuels with the control of greenhouse-gas emissions and associated environmental consequences. The carbon dioxide is stored as supercritical, gaseous, dissolved in formation water or converted into solid carbonates, depending on the lithology of the host rock, the targeted depths and associated pressures and temperatures. In recent years, subsurface storage into basalts or peridotites, rich in calcic and ferromagnesian silicates, has been considered because these rocks have high potential for secure and long-term carbonation[1,2]. Through the dissolution of their silicate components, (ultra)mafic rocks have the potential to trap, as precipitated Ca-Mg-Fe-carbonates, significant quantities of $CO_2$ in Earth's crust.

Field and experimental work has been performed to understand the physico-chemical mechanisms governing carbonation reactions in basalt and to optimize associated rates. Until now, the deep biological component present at storage sites has rarely been considered[3]. Most of the work involving microbiology in the carbon capture and storage (CCS) field has targeted sedimentary basins[4–8] where carbonation is negligible due to the lack of reactive minerals. $CO_2$ injections constitute an oxidative and acidifying perturbation, however, these studies have shown that the injected $CO_2$ represents a valuable carbon source for life with the potential to enhance chemolithoautotrophic activities[7,8]. Conversely, little is known about the nature and metabolic diversity of microbial communities living in basalt-systems[9] or the biogeochemical reactions they can provoke following $CO_2$ injection, including biologically-enhanced or -inhibited conversion of $CO_2$ into solid carbonates[10]. Their ability to impact the chemistry of their environment and subsequently affect the fate of injected $CO_2$ makes deep-indigenous microbial populations potential critical factors for carbon immobilization.

Our study focuses on the $CO_2$-injection site associated with the Hellisheidi geothermal powerplant (SW-Iceland) and developed in the framework of the Carbfix project to assess the feasibility of carbon capture and in situ mineral storage in basalt[2,11]. Our objective was to characterize the nature and reactivity of the microbial community hosted in the storage formation groundwater. We assess species richness, phylogenetic diversity and abundance along with metabolic potential by bacterial 16S-rRNA gene 454-pyrosequencing, as well as archaeal and bacterial 16S-rRNA gene Sanger sequencing, quantitative polymerase chain reaction (qPCR), PCR detection of key genes for inorganic carbon assimilation and degradation of aromatic compounds and metagenomic analysis. These approaches allow us to enumerate function and identity genes, and evidence changes compared to the native microbial community.

## Results

**Gas injections**. The storage formation, located in the 400–800 m depth interval, is a lava-flow sequence composed of weakly altered crystalline basalts of olivine-tholeiitic composition with temperatures ranging between 20–50 °C[11] (Supplementary Fig. 1). From mid-January to August 2012, 175 t of commercial $CO_2$ and 73 t of a gas mixture, derived from the purification of the geothermal gas harnessed by the plant (75% $CO_2$-24.2% $H_2S$-0.8% $H_2$), were consecutively injected with reactive and non-reactive tracers (Supplementary Fig. 2; Supplementary Table 1)[12–14]. In injection well HN-02, the gas was mixed at 350 m depth with groundwater pumped from control well HN-01 (Supplementary Fig. 1)[2]. This mixing produced an acidic water (pH ~ 3.9–4.0) with a concentration of dissolved $CO_2$ of $820 \pm 41$ mmol l⁻¹ for the pure-$CO_2$ injection and

$430 \pm 22$ mmol l⁻¹ for the mixed-gas injection[13–15]. Groundwater was sampled before and during gas injections from both control well HN-01 and monitoring well HN-04 (Supplementary Figs. 1 and 2). Changes in HN-04 groundwater community are discussed with respect that of HN-01, which was unaffected by the $CO_2$-plume (Methods; Supplementary Fig. 1). Previous studies have shown that 95% of the injected $CO_2$ expected to reach the first monitoring well, HN-04, between 200 to 400 days after injection onset, was instead immobilized at depth upstream to HN-04[13]. This study focuses on the fast-flow pathway (i.e., fractures and rubbles) through which a few percent of the gas-charged water travelled during the two first months after the injection (i.e., from February to May 2012; Supplementary Fig. 2)[13,16]. This pathway represents a small portion of the injected gas, but it is the largest $CO_2$ fraction that reached monitoring well HN-04[13,14]. From this fraction it was possible to assess the reactivity of the aquifer's microbial inhabitants in response to $CO_2$-enriched groundwater.

**Host-rock dissolution upon $CO_2$ acidification**. The fast-flowing fraction of the pure-$CO_2$ charged groundwater was first detected at HN-04 in February 2012 but the most elevated dissolved inorganic carbon (DIC) concentrations were reported in March 2012, ~ 60 days after the beginning of the injection (Supplementary Fig. 2a). During that time, DIC concentration increased from an initial value of $1.5 \pm 0.1$ to $4.4 \pm 0.2$ mmol l⁻¹ (ref. 12). Groundwater pH dropped first from $9.6 \pm 0.1$ to $9.1 \pm 0.1$ in February 2012, then down to $7.3 \pm 0.1$ by early March 2012 (Fig. 1a), with minimal values of 6.6 mid-March[12–14] (Supplementary Fig. 2b). Opposite variations for conductivity were simultaneously observed, reaching $313 \pm 3$ μS cm⁻¹ in March 2012 against $249–263 \pm 3$ μS cm⁻¹ before gas injections (Fig. 1b). The concomitant decrease in pH and increase in conductivity directly resulted from acidification by the $CO_2$-rich groundwater that induced host-rock dissolution and release of ions (notably $Fe^{2+}$, $Ca^{2+}$, $Mg^{2+}$, $Zn^{2+}$; Supplementary Tables 2 and 3; Supplementary Fig. 2d). In May 2012, pH and conductivity were near their original values but they fluctuated once again along with metals concentrations in July 2012 with the arrival of the portion of geothermal gas mixture transported along the fast-flow pathway (Supplementary Fig. 2a, b, d, e; Supplementary Table 3). pH and conductivity varied only weakly upon injection in groundwater of control well HN-01 (Fig. 1a, b).

Na-fluorescein (Na-flu) inherited from former hydrological tests (Supplementary Table 1) and slowly transported by matrix flow[16] was recovered in HN-04 groundwater during the monitoring period. Na-Flu concentrations increased slightly in October 2010 up to $2.82 \cdot 10^{-6}$ g l⁻¹ and then during the next sampling periods where their values were increased by a factor ~ 20 (maximal values of $5.54 \cdot 10^{-5}$ g l⁻¹; Supplementary Fig. 2c). Dissolved organic fragments, detected by Fourier transform ion cyclotron resonance mass spectrometer (FT-ICR-MS) and expressed as CHO, CHOS, CHON, CHONS molecular series, also showed an increase in their respective diversity with their maximal values observed in May 2012 (Supplementary Figs. 3 and 4).

**Initial bacterial community**. Pyrosequencing analyses showed that before the injections, and similar to HN-01, the HN-04 bacterial community was diverse with a number of operational taxonomic units (OTUs) ranging from 274 to 421 (Fig. 1c; Supplementary Figs. 5 and 6). It was mainly composed of Proteobacteria, Nitrospirae, and Chlorobi (Figs. 2 and 3). In July 2009, the dominant species belonged to the alphaproteobacterial Sphingomonadales order (22.7–23.2% of the pyrosequences;

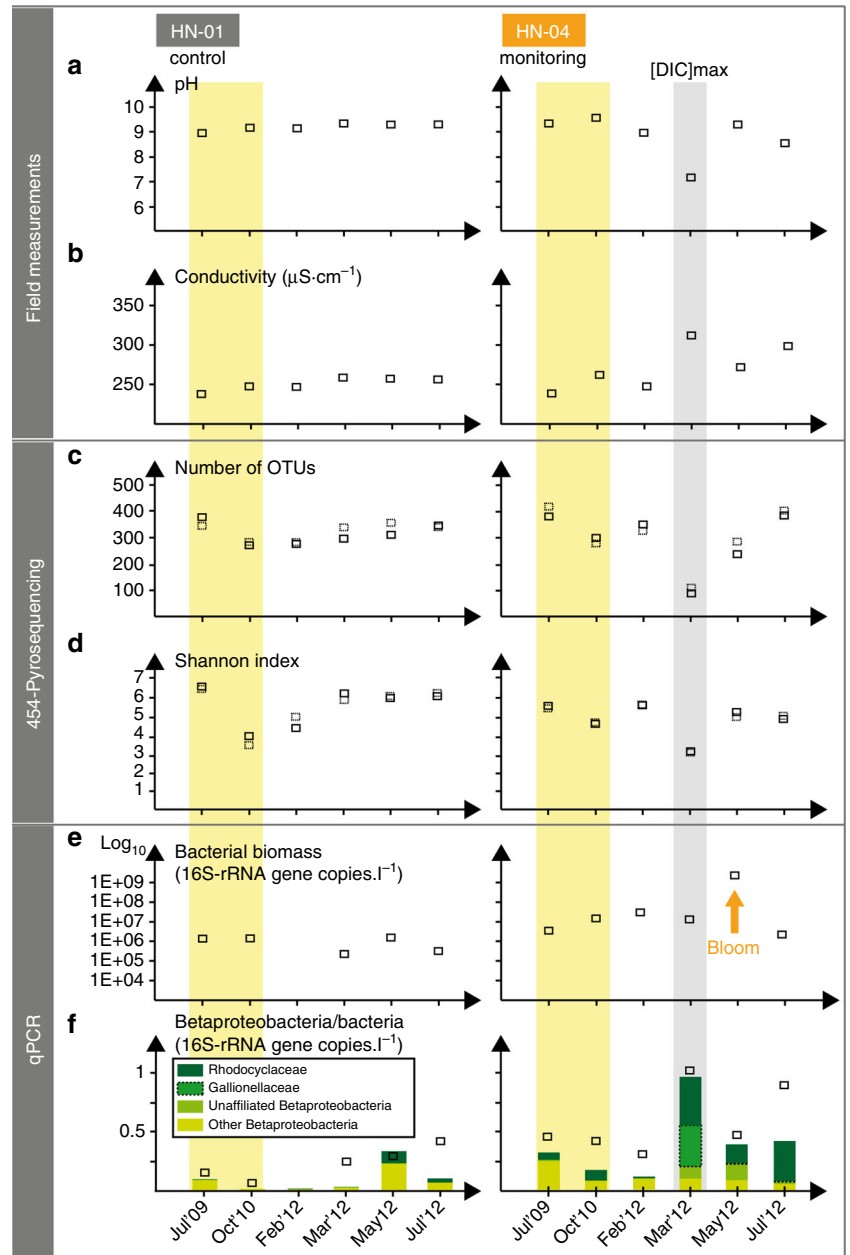

**Fig. 1** Contrasting temporal patterns in groundwater of monitoring well HN-04 and control well HN-01. pH (**a**) and conductivity (**b**) measured in the field, number of operational taxonomic units (OTUs) (**c**) and Shannon index (**d**), both obtained in duplicates from 454-pyrosequencing of 16S-rRNA amplicons (Supplementary Fig. 5). Also shown is gene abundance obtained using quantitative polymerase chain reaction (qPCR) for bacteria (**e**) and Betaproteobacteria (**f**). Betaproteobacteria taxonomy established from pyrotag sequencing is also indicated in **f**. The yellow boxes frame the pre-injection measurements, whereas the gray box indicates the time period displaying the highest dissolved inorganic carbon (DIC) concentration at the level of well HN-04 (March 2012) that resulted in groundwater acidification shown in **a**. Errors (i.e., standard deviation) on analytical measurement for pH and conductivity are smaller than symbols. For details on the injection site and geochemical monitoring, see Supplementary Figs. 1 and 2, respectively, along with Supplementary Tables 1 to 3

Fig. 3) whose members are mainly aerobic chemoheterotrophs[17]. In October 2010, bacteria belonging to order Ignavibacteriales became dominant (43.1–45.4%). Cultivated representatives of this order are capable of organoheterotrophy under both oxic and anoxic conditions[18]. The differences in bacterial composition observed before gas injection in HN-04 could be related to fluctuating concentrations of dissolved $O_2$ ($4 \pm 1$ to $14 \pm 1 \, \mu mol \, l^{-1}$, although atmospheric contamination cannot be excluded for samples containing little oxygen[14]) and increased $NO_3^-$ concentrations in July 2009 ($2.465 \, \mu mol \, l^{-1}$ vs. pre-injection mean value of $0.463 \, \mu mol \, l^{-1}$ ($\sigma = 0.716$); Supplementary Table 2). No clear

autotrophic groups of bacteria were detected before gas injection although DIC concentrations were ~ 50 to 90 times higher than that of dissolved organic carbon (DOC) in October 2010 and July 2009, respectively ($1.5$ vs. $0.03 \, mmol \, l^{-1}$ and $1.9$ vs. $0.02 \, mmol \, l^{-1}$ for DIC and DOC, respectively; Supplementary Table 2).

In early February 2012, changes in HN-04 groundwater chemistry were observed accompanying the pH drop of ~ 0.6 units (Fig. 1a; Supplementary Table 3). The planktonic microbial community inhabiting the aquifer close to well HN-04 were not strongly impacted by $CO_2$ injection based on the number of OTUs, the Shannon index, and the quantity of bacteria that all

remained relatively stable (Fig. 1c–e). Changes, including the increase in the proportion of OD1 Parcubacteria, stayed in the range of the aquifer natural dynamics as observed for HN-01 (Fig. 4). Nonetheless, among all the primers tested for PCR amplification of genes involved in C-fixation (Supplementary Table 4), we began to detect *cbbL* genes of Gamma- and Betaproteobacteria (Supplementary Fig. 7). Those genes encode the form I ribulose-1,5-bisphosphate carboxylase/oxygenase (RuBisCO), a key enzyme for autotrophic $CO_2$-fixation in the Calvin-Benson-Bassham (CBB) cycle[19]. *cbbL* cycle genes were not previously detected in HN-04 and were never amplified from HN-01 groundwater during the whole survey, suggesting that $CO_2$ fixing bacteria were specifically associated with the arrival of the $CO_2$-charged groundwater.

**Community associated with $CO_2$-charged groundwater.** The main arrival of the fast-flowing fraction of the pure-$CO_2$ charged groundwater to HN-04 in March 2012 was observed with an increase in DIC (Supplementary Fig. 2a), a decrease of pH by ~ 2 units (Fig. 1a; Supplementary Fig. 2b), an increase in free ion concentrations (Supplementary Fig. 2d; Supplementary Table 3), and a change in conductivity ($263 \pm 3$ to $313 \pm 3\,\mu S\,cm^{-1}$; Fig. 1b)[12–14]. Changes were also observed in the microbial population (Figs. 1–4; Supplementary Figs. 6 and 10). The number of OTUs retrieved from HN-04 groundwater decreased strongly from 323–354 to 89–107 (Fig. 1c). However, the total amount of 16S-rRNA gene copies remained constant (Fig. 1e). Betaproteobacteria, already present in low abundance prior to injection, became dominant after, accounting for 87.5–87.8% and 92% of the 16S-rRNA gene sequences obtained by pyrosequencing and metagenomic analysis, respectively (Figs. 1f, 2b, and 3; Supplementary Fig. 10). Decrease in pH and rise in DIC and metal concentrations (Fig. 1a; Supplementary Fig. 2a; Supplementary Table 3) fostered proliferation of this population as supported by statistical analysis (Fig. 5; Supplementary Table 5) showing in addition significant Pearson correlations between Betaproteobacteria and pH ($r = -0.90$, $p$-value = 0.010), DIC ($r = 0.97$, $p$-value = 0.006) and $Fe^{2+}$ concentrations ($r = 0.92$, $p$-value = 0.009).

Among Betaproteobacteria, the Gallionellaceae family, only weakly detected before injection (<1% of the pyrosequences), became the dominant group in March 2012 (34.7–35.5% of the pyrosequences and 40% of the 16S-rRNA gene sequences retrieved by metagenomic analysis; Figs. 1f, 2b, and 3; Supplementary Fig. 10). Most of them were affiliated to *Sideroxydans lithotrophicus* species (DQ386859 and CP001965; Supplementary Table 6; Supplementary Fig. 10). All cultivated members of the Gallionellaceae family characterized so far are chemolithoautotrophic microaerophilic iron-oxidizing bacteria (FeOB) who grow at circumneutral or acidic pH[20,21]. The microoxic conditions of the aquifer (Supplementary Table 2), along with the increase of DIC and $Fe^{2+}$ concentrations and a pH

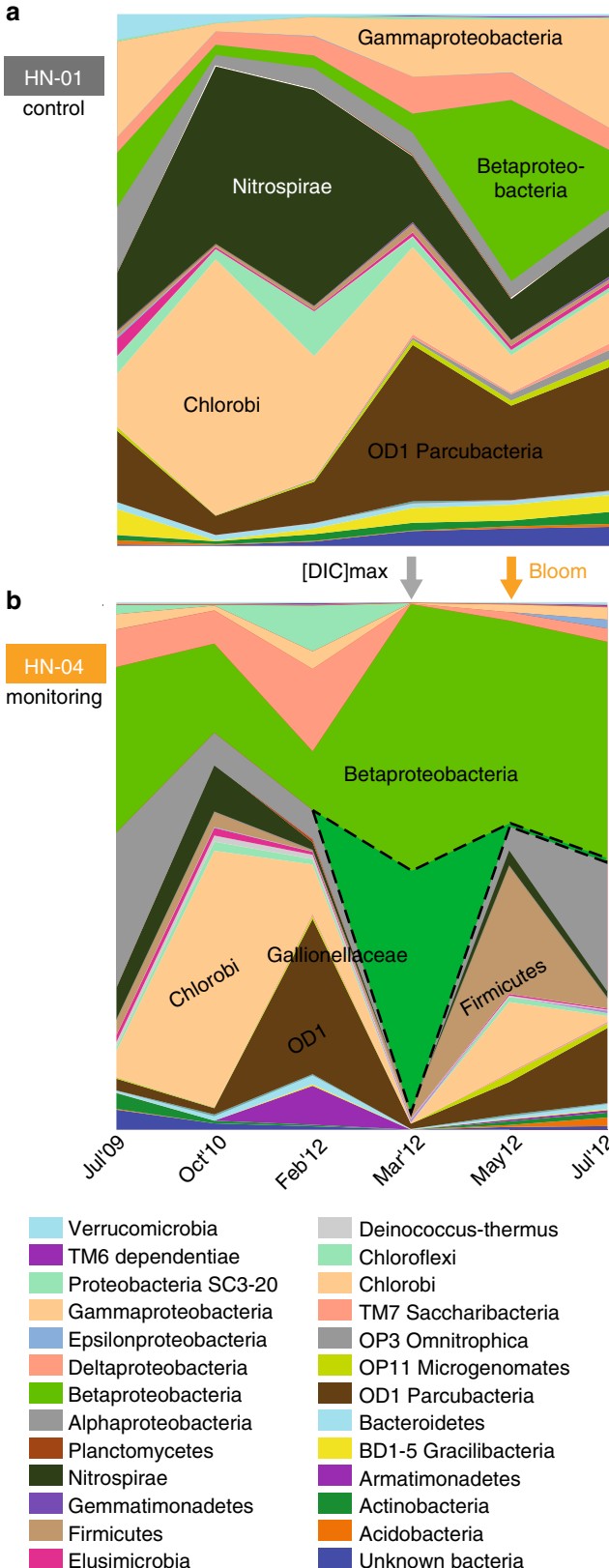

**Fig. 2** Post-injection bacterial blooms based on taxonomic distribution of the 16S-rRNA gene community profile obtained from 454-pyrosequencing. The post-injection bacterial blooms are represented as a function of time in the groundwater of control well HN-01 (**a**) and monitoring well HN-04 (**b**), with colors representing different taxa. Groundwater of control well HN-01 exhibited a relatively stable bacterial community with Chlorobi, Nitrospirae, OD1 Parcubacteria and Proteobacteria as dominant community members. While broadly similar to the pre-injection community sampled from HN-01 groundwater, the HN-04 community structure was consistently different after the pure-$CO_2$ injection. In particular, growth of Betaproteobacteria accounting for 88% of the community in March 2012, was first favored; note a bloom of Gallionellaceae related species, which were only weakly detected before the injection (similarly shown by metagenomic analysis; Supplementary Fig. 10). It was followed by Firmicutes accounting for more than 20% of the groundwater community in May 2012, hence dominating with betaproteobacterial *Thiobacillus* species (Supplementary Fig. 10)

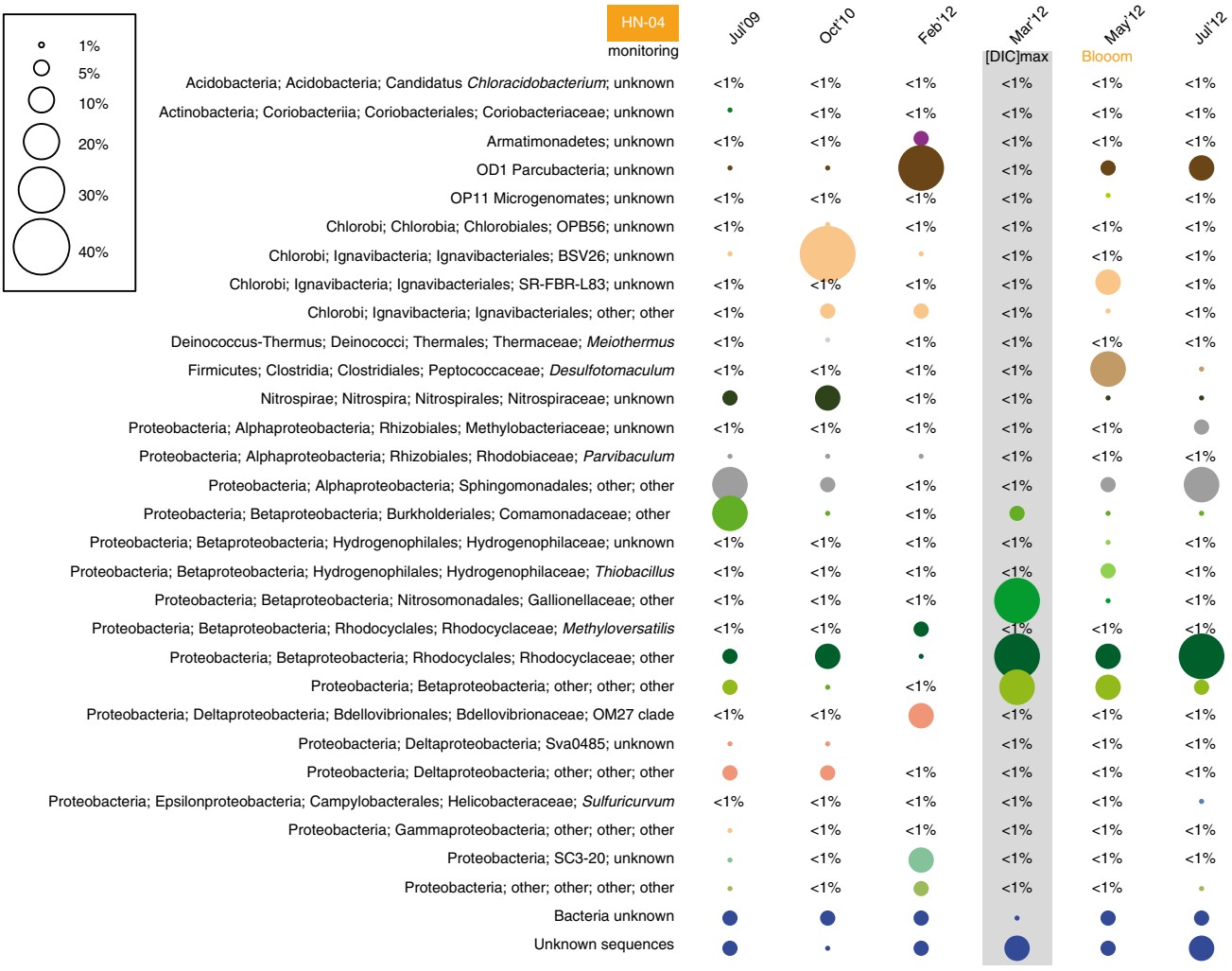

**Fig. 3** Relative abundance of OTUs retrieved by 454-pyrosequencing of the 16S-rRNA genes in HN-04 groundwater for the six sampling periods. OTUs are defined at a sequence similarity ≥97%. OTUs retrieved at less than 1% during any time period are not represented. For each OTU, the size of the circle represents the exact percentage of its abundance. In the size scale, some reference points are given for abundances ranging from 1 to 40%. The gray box indicates the sampling period displaying the highest DIC concentration for well HN-04 groundwater

around 7 as observed in March 2012 for HN-04 groundwater (Fig. 1a; Supplementary Fig. 2a, b, d; Supplementary Table 3) created a favorable ecological niche for the development of these iron-oxidizing Betaproteobacteria, as highlighted by statistical analysis showing significant linear correlations of Gallionellaceae and *Sideroxydans* sp. with pH ($r = -0.93$, $p$-value $= 0.008$; $r = -0.92$, $p$-value $= 0.009$, respectively), DIC ($r = 0.95$, $p$-value $= 0.010$; $r = 0.94$, $p$-value $= 0.020$, respectively) and $Fe^{2+}$ concentrations ($r = 0.99$, $p$-value $< 0.001$; $r = 0.99$, $p$-value $< 0.001$, respectively). In the metagenomic data we detected genes coding for various cytochromes involved in Fe-oxidation pathways including *cytC552* and *mtrA* (Fig. 6; Supplementary Table 7). One homolog of the *mtrA* gene, *mtoA*, encodes an outer membrane *c*-type cytochrome putatively responsible for Fe-oxidation in *S. lithotrophicus*[20,22]. In addition, matching our results to the Pfam database[23], we found, among the most abundant protein-sequence motifs retrieved in March 2012, several genes involved in the biogenesis of *c*-type cytochromes, which are related to iron metabolisms (Supplementary Fig. 11). Remarkably, the second most abundant protein-sequence motif coded for the high-potential iron-sulfur protein that was identified as the iron-oxidizing protein involved in several strains related to *Acidithiobacillus*[24]. The most abundantly retrieved protein-sequence motif

was PhnJ, an ABC transporter for phosphonate uptake. Increased expression of phosphate-acquiring transporters is in favor of effective microbial iron oxidation in the aquifer. High insolubility of the produced ferric iron leads to the precipitation of biogenic iron oxides to which phosphate strongly binds[25]. PhnJ is nonetheless not specific to Gallionellaceae and has been found only in acidophilic *Sideroxydans* strains[21]. This agrees with canonical correlations, which suggest the community present in March 2012 in the aquifer may have a higher tolerance to lower pH (Fig. 5). Although sequences are lacking in the COG[26] and Pfam[23] databases, we detected in the metagenomic data set homologs of the *cyc2* gene present in *S. lithotrophicus* that were 19 times more abundant in March 2012 than in May 2012 (Supplementary Table 8). Homologs of this gene potentially coding for an outer membrane Fe-oxidizing *c*-type cytochrome have been found in all the genomes sequenced so far of neutrophilic microaerophilic FeOB, as well as in metagenomes containing Gallionellales and Zetaproteobacteria members[27]. Furthermore, evidence for abundant expression of a *cyc2* homolog was recently found in an aquifer where Gallionellales were present[28]. Markers for sulfur and sulfite oxidation (*soxYZ*) and dissimilatory sulfate and sulfite reduction and oxidation (*dsrAB*, *aprAB*) were also detected by metagenomic analysis,

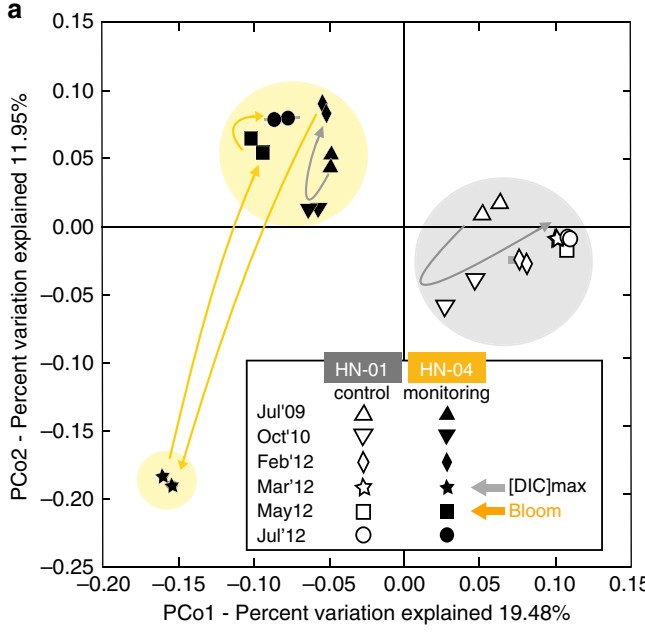

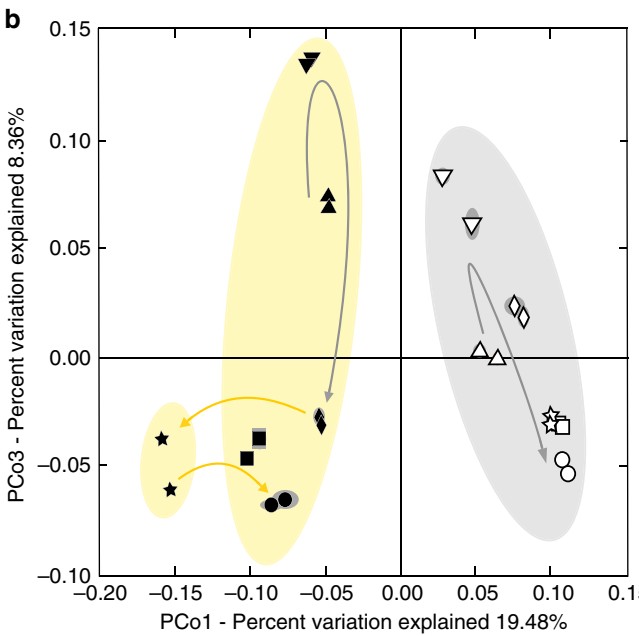

**Fig. 4** Principal coordinates analysis of the relative abundance of the 16S-rRNA genes retrieved by 454-pyrosequencing. It reflects injection-induced shifts in the composition of the groundwater bacterial community of monitoring well HN-04 (filled symbols), compared to control well HN-01 (hollow symbols). The percentage of variations explained by the first three principal coordinates is indicated on the axes (in **a** and **b** for PCo1/PCo2 and PCo1/PCo3, respectively). Lines and halos on symbols represent numerical uncertainties (1,000 replicates). Arrows illustrate evolution through time. A continuous trend characterizes the HN-01 bacterial community, likely reflecting natural variations of the aquifer chemistry[11] (Supplementary Table 2). Chemical variability is naturally induced by regional groundwater flow along with magmatic degassing from the Hengill volcanic system. Although the drift is also visible in well HN-04 groundwater, its evolution is typified by a disruptive and rapid shift upon acidifying $CO_2$ injection, which drastically impacted the community structure in March 2012

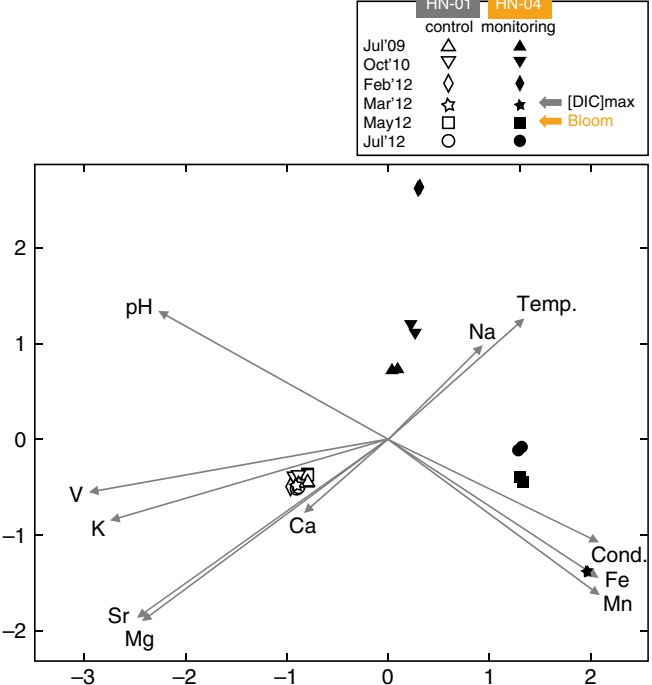

**Fig. 5** Canonical correlations based on variance analysis all along the microbiological survey. The survey was carried out between July 2009 and July 2012. The represented geochemical parameters are variables (from Supplementary Tables 2 and 3) whose variance is correlated with the OTU relative abundances obtained from 454-pyrosequencing of 16S-rRNA amplicons (Figs. 2, 3, and Supplementary Fig. 5). The longer the arrow, the stronger the correlation. Monitoring well HN-04 features a strong correlation between Fe and Mn concentrations in groundwater and the OTUs retrieved in March 2012 (and to a lesser extend in May 2012 and July 2012)

agreeing with the presence of a *soxXYZAB* cluster and Dsr and Apr complexes in *S. lithotrophicus* ES-1 genome[20].

The bloom of Gallionellaceae members is indicative of an increased potential for autotrophic C-fixation and thereby $CO_2$ conversion into biomass in the aquifer in March 2012. This is further supported by selective amplifications in addition to *cbbL* genes of *cbbM* genes encoding form II RuBisCO[19] (Supplementary Figs. 7 and 8; Supplementary Table 4). *cbbM* genes, adapted to functioning under low $O_2$ and high $CO_2$ concentrations[29], were only detected in March 2012 when DIC concentrations were the most elevated. Some of the *cbbM* gene sequences closely related to environmental sequences of this gene found in iron-rich freshwater (AB722270.1). One sequence closely matched with the *cbbM* gene of *S. lithotrophicus* (CP001965.1), in agreement with the main affiliation found by 16S-rRNA gene analysis for the bloom of Gallionellaceae (Supplementary Fig. 10; Supplementary Table 6). The presence of the CBB cycle was confirmed by metagenomic analysis. We detected genes coding for phosphoribulokinase (*prkB*) and subunits of ribulose-1,5-bisphosphate carboxylase (*rbcL*, *rbcS*) (Fig. 6; Supplementary Table 7). No other $CO_2$-fixation pathways were detected. Autotrophic pathways were further supported by the high level of *nifD* markers retrieved in March 2012 (Fig. 6). This gene, present in *S. lithotrophicus* genome[20,30], codes for a key enzyme involved in nitrogen-fixation.

In contrast with the Gallionellaceae, the other blooming Betaproteobacteria (31.0–31.6% Rhodocyclaceae, 3.0–3.1% Comamonadaceae, 17.1–18.9% unaffiliated, based on pyrose-quencing) were also abundant before the injection (Fig. 3).

Metagenomic analyses confirmed the dominance of Rhodocyclaceae (30%), with the remaining fractions distributed between Comamonadaceae (13%), Hydrogenophilaceae (6%), Burkholderiaceae (2%), Nitrosomonadaceae (1%) (Supplementary Fig. 10). Compared to the Gallionellaceae, the potential metabolic capabilities of these Betaproteobacteria were difficult to assess as they all belong to versatile groups. The diversity of *cbbL* and *cbbM* genes showed that they were closely related to genes of cultivated betaproteobacterial representatives of the Comamonadaceae and Burkholderiaceae families. This suggests that the Gallionellaceae were not the only Betaproteobacteria able to fix $CO_2$ (Supplementary Figs. 7 and 8). However, concomitantly with the appearance of the unaffiliated Betaproteobacteria group in March 2012 (as highlighted by pyrosequencing) and its persistence in May 2012, we detected using PCR amplifications, sequences related to betaproteobacterial genes coding for the largest subunit of multicomponent phenol hydroxylases (LmPHs; Supplementary Table 4) involved in the degradation of phenolic compounds[31] (Supplementary Fig. 9). These genes were also closely related to genes of cultivated betaproteobacterial representatives of the Rhodocyclaceae, Comamonadaceae, and Burkholderiaceae families. Metagenomic analysis also showed abundant key markers for several pathways of aerobic degradation of aromatic compounds with the most prominent potential marker coding for 2-polyprenyl-6-methoxyphenol hydroxylase and related FAD-dependent oxidoreductases (Fig. 6; Supplementary Table 7). Genes encoding several Fe-dependent aromatic ring-opening dioxygenases (*ligB*, *gloA*), as well as genes coding for Zn-dependent dienelactone hydrolase (DLH) and putative aryl-alcohol dehydrogenase (*tas*) were detected together with markers for anaerobic degradation of aromatic compounds (*bcrC/badD/hgdB*). Abundant markers for fermentation were also highlighted (*porA-G*, *pta*, *ackA*). Therefore, some of the represented Betaproteobacteria were likely fermentative bacteria or degraders of aromatic compounds. Additionally, markers for denitrification (*nirK*, *nirS*, *nirB*, *nosZ*) were detected suggesting that some used nitrate as electron acceptor.

### General bloom of subsurface bacteria following $CO_2$ injection.
In May 2012, DIC concentrations decreased in HN-04 groundwater when compared to March 2012, but values still reached 2.5 ± 0.1 mmol $l^{-1}$ (Supplementary Fig. 2a). Na-flu concentrations remained elevated (4.96·$10^{-5}$ g $l^{-1}$; Supplementary Fig. 2c), while pH, rapidly buffered by the host–rock[11–14], returned to its original value of 9.1 (Fig. 1a). As shown by the Shannon index

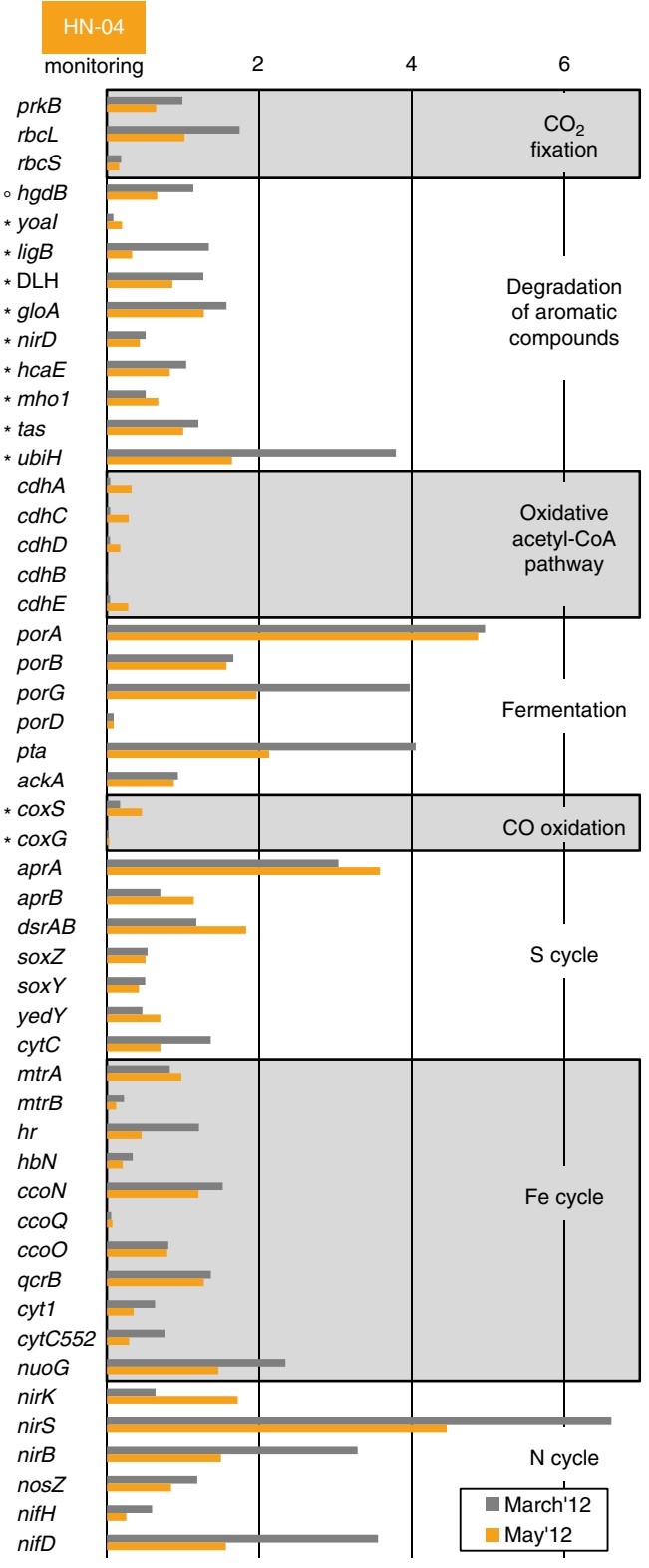

**Fig. 6** Normalized counts of key biomarkers in groundwater from monitoring well HN-04 sampled in March and May 2012. *prKB* phosphoribulokinase, *rbcL-S* ribulose 1,5-bisphosphate carboxylase, large and small subunits, *hgdB* benzoyl-CoA reductase, *yoaI* aromatic ring hydroxylase, *ligB* aromatic ring-opening dioxygenase, DLH dienelactone hydrolase, *gloA* catechol 2,3-dioxygenase, *nirD* ring-hydroxylating dioxygenase, *hcaE* phenylpropionate dioxygenase or related ring-hydroxylating dioxygenase, *mhol* predicted class III extradiol dioxygenase, *tas* predicted oxidoreductase, *ubiH* 2-polyprenyl-6-methoxyphenol hydroxylase, *cdhA-E* CO dehydrogenase/acetyl-CoA synthase subunits, *porA-D* pyruvate:ferredoxin oxidoreductase or related 2-oxoacid:ferredoxin oxidoreductase subunits, *pta* phosphotransacetylase, *ackA* acetate kinase, *coxS-G* aerobic-type carbon monoxide dehydrogenase subunits, *aprA-B* adenylyl sulfate reductase, *dsrAB* sulfite reductase, dissimilatory type, *soxZ-Y* sulfur oxidation protein, *yedY* sulfite oxidase, *cytC* sulfite dehydrogenase, *mtrA* MR-1 decaheme cytochrome, *mtrB* decaheme cytochrome, *hr* hemerythrin, *hbN* truncated hemoglobin, *ccoN-O* cbb3-type cytochrome oxidase subunits, *qcrB* cytochrome b subunit of the bc complex, *cyt1* cytochrome c1, *cytC552* cytochrome c551/c552, *nuoG* NADH dehydrogenase/NADH/ubiquinone oxidoreductase, *nirK* copper containing nitrite reductase, *nirS* cytochrome cd1 containing nitrite reductase, *nirB* Nad (P)H-nitrite reductase, *nosZ* nitrous oxide reductase, *nifH-D* nitrogenase subunits. When relevant, aerobic and anaerobic pathways are indicated by a circle and a star, respectively (see Supplementary Table 7). Note that *ubiH* could also be involved in the synthesis of ubiquinone used in conjunction with the $bc_1$ cytochrome for iron oxidation in *Sideroxydans* sp[22]

(Fig. 1d), bacterial diversity recovered to its initial levels. However, a factor of 500 increase of the bacterial biomass has been estimated based on 16S-rRNA gene copy numbers (Fig. 1e). This agrees with the highest diversity of organic fragments inherited from fatty acid (lipid-like) and protein-like structures as observed with FT-ICR-MS (Supplementary Fig. 3). The increased diversity of nitrogen-bearing compounds (CHON, CHONS molecular series) can be related to an abundance of peptides in the screened mass domain (Supplementary Fig. 4), in favor of enhanced biological activity. It has been speculated that biofilms sheared from host-rock or well pipe can account for shifts in microbial diversity and abundance in groundwater experiments[32]. We controlled for such shearing by maintaining the same sampling protocol in all wells over all sampling periods (Supplementary Fig. 12; Methods). No other observations from HN-04, or 5 other monitoring wells, reflect a similar increase in biomass. Therefore, the observed bloom likely relates to favorable growth conditions resulting from geochemical changes induced by the circulation of pure-$CO_2$ charged groundwater. During this period increased concentrations, compared against pre-injection, of metals and DIC are observed along with a return to the baseline pH (Fig. 1a; Supplementary Fig. 2a, d; Supplementary Tables 2 and 3).

During May 2012, Gallionellaceae were weakly retrieved (<1% of pyrosequences and 2% for metagenomic analysis) but markers of the CBB C-fixation pathway and markers for sulfur-and iron-oxidation pathways were still detected (Fig. 6, Supplementary Fig. 11; Supplementary Table 7). As in March 2012, no other $CO_2$-fixation pathways were detected. In May 2012, metagenomic data showed the presence of facultatively-anaerobic Betaproteobacteria belonging to the *Thiobacillus* genus whose proliferation was favored compared to the Gallionellaceae (26% of 16S-rRNA gene sequence in metagenomic data and 3.1–3.6% of the pyrosequences). Sanger sequencing of the 16S-rRNA encoding genes that allows establishing more robust phylogenetic affiliation, showed that sequences shared 99% of identity with the autotrophic strain *Thiobacillus denitrificans* (CP000116; Supplementary Table 6). This strain, known to oxidize sulfur and iron species, has the capability to grow by reducing nitrate and nitrite[33]. This agrees with the metagenomic analyses highlighting denitrification and Fe/S-oxidation markers (Fig. 6; Supplementary Fig. 11; Table 7). The similar proportions of iron- and sulfur-oxidation markers retrieved in March and May 2012 could be explained by the significant gene homology and gene order shared by the *soxXYAB* gene clusters of *T. denitrificans* and *S. lithotrophicus* species[20]. However, in May 2012 the most abundant protein-sequence motifs relating to iron metabolisms differed from those retrieved in March 2012, suggesting a change in the Fe/S-oxidation pathways used during the two blooms (Supplementary Fig. 11).

Representatives of the Firmicutes phylum affiliated to the *Desulfotomaculum* genus became dominant in May 2012 (16.4–23.2% of the pyrosequences and 12% of the metagenomic data; Fig. 3; Supplementary Fig. 10) after only being weakly detected pre-injection (<1% of the pyrosequences). The presence of this phylum in conjunction with denitrifying bacteria suggests the existence in the aquifer of anoxic zones compatible with the development of strict anaerobes around the May 2012 sampling period. *Desulfotomaculum* species are known to oxidize organic compounds while reducing sulfur species[34]. *Desulfotomaculum profundi*, the closest cultivable relative to the 16S-rRNA gene sequence retrieved in May 2012 (Supplementary Table 6), was isolated from a community capable of degrading ethylbenzene, toluene, and benzene[35]. In May 2012, metagenomic data showed an increase of sulfate reductases of dissimilatory type (*dsrAB*) (Fig. 6; Supplementary Table 7). Additionally, iron-dependent hydrogenase, present in sulfate reducers[36], was the most abundant iron-dependent protein-sequence motif retrieved for May 2012 (Supplementary Fig. 11). We also observed an increase of CO dehydrogenase/acetyl-CoA synthase (*cdh*) subunits involved in the oxidative Acetyl-CoA pathway, a pathway characteristic to strict anaerobes degrading aromatic rings. This yields Acetyl-CoA, which is subsequently oxidized to $CO_2$ fueling the activation steps of rings destabilization[37]. Therefore, the *Desulfotomaculum*-related OTU dominant in May 2012 may correspond to degraders of aromatic cycles. At the same time, pyrosequencing and metagenomic analysis showed that Rhodocyclaceae were still dominant (respectively representing 12.2–12.9% of the pyrosequences and 27% of the 16S-rRNA gene sequences in the metagenomic data) and that Ignavibacteriales, Sphingomonadales and OD1 Parcubacteria, originally inhabiting the aquifer, were retrieved again (representing, respectively, 11.0–12.4%, 2.1–3.7% and 5.3–5.4% of the pyrosequences and 6, 7 and 1% of the 16S-rRNA gene sequences in the metagenomic data).

In July 2012, temporal evolutions of non-reactive $SF_5CF_3$ tracer, DIC, and $H_2S$ concentrations indicated that the fast-flowing fraction of the geothermal gas mixture had reached HN-04 (Supplementary Fig. 2a, e; Supplementary Table 1). This was confirmed by a pH decrease (~ 1 unit; Fig. 1a; Supplementary Fig. 2b), and an increase of both conductivity (Fig. 1b) and metal concentrations (Supplementary Fig. 2d; Supplementary Table 3) in HN-04 groundwater. The DIC concentrations increased slightly to $2.5 \pm 0.1$ mmol l$^{-1}$ and the concentration of Na-fluorescein decreased against May 2012 values ($4.01 \cdot 10^{-5}$ g l$^{-1}$; Supplementary Fig. 2c). The bacterial community was dominated by Rhodocyclaceae (27.8–29.5% of the pyrosequences) and Sphingomonadales (16.1–17.5%). We did not detect any 16S-rRNA genes clearly suggesting the presence of autotrophic bacteria, neither did we detect any RuBisCO encoding genes by PCR amplification. This suggests that the changing environmental parameters observed in July 2012 were not in favorable for autotrophic growth. Such growth may have been limited by pH, $H_2S$, $O_2$, and nitrogen species' concentrations.

Based on metagenomic analysis we found that the majority of the observed ecosystems during the injection survey were made up of bacteria (Supplementary Fig. 10). Eukaryotes, potentially fed by the bacterial bloom, were slightly more abundant in May 2012 (2% of the 16S-rRNA gene sequences). They were mainly constituted by Euglenozoa. Archaea were retrieved at similar level (2%) in March and May 2012. Only two archaeal species were detected in HN-04 groundwater during the whole survey, belonging to the Thaumarchaeota phylum and the Terrestrial Miscellaneous Euryarchaeotal Group (Supplementary Figs. 13 and 14).

## Discussion

Only a small fraction of the injected gas reached HN-04 monitoring well through fast-flow pathway, however, we observed rapid and large changes in HN-04 groundwater microbial community in response to pure-$CO_2$ injection. The microbial response to the arrival of $CO_2$-charged water trends with DIC evolution and groundwater geochemical data[12–14], as supported by statistical analysis (Fig. 5; Supplementary Fig. 2; Supplementary Table 5). Successional microbial dynamics were observed in the deep aquifer, and bacteria belonging to the betaproteobacterial class were most fostered by the changes in aquifer chemistry. The gas injection altered geochemical conditions by introducing acidified waters, which prompted rock dissolution resulting in ion release[14] (Supplementary Table 3). The release of ions may have been beneficial for certain bacterial types as suggested by Mantel correlations (Fig. 5; Supplementary Table 5). Fe$^2$

+ likely constituted an energy source for $CO_2$-assimilating FeOB belonging to the Gallionellaceae family (detected in March 2012) and the *Thiobacillus* species (detected in May 2012). As supported by Pearson correlations, the decrease in the Gallionellaceae number of representatives in May 2012 is likely related to pH ($r = -0.93$, *p*-value = 0.008), $Fe^{2+}$ ($r = 0.99$, *p*-value < 0.001) or oxygen concentrations no longer compatible with their development (Figs. 1a and 5; Supplementary Fig. 2d; Supplementary Table 3). In addition, Fe, Zn, Mn, and Mg are needed as cofactor in many of the enzymes highlighted by metagenomic analysis (Supplementary Table 7). Their release from the host basalt was likely beneficial for both heterotrophs and autotrophs. Most markers of the degradation of aromatic compounds dependent on Zn (DLH, *tas*), Fe (*ligB*, *gloA*, *nirD*, *hcaE*), Mn (*hgdB*) and Mg (*hgdB*) (Fig. 6) or alphaproteobacterial Fe/Zn-bearing LmPHs encoding genes (Supplementary Fig. 9) were more abundant in March 2012 when Fe, Zn, Mg concentrations were more elevated in HN-04 groundwater (Supplementary Table 3). Similarly, the growth of *Desulfotomaculum* species that became dominant in May 2012 need $Ca^{2+}$ (ref. 34), whereas $Mg^{2+}$ is mandatory for the activation of the RuBisCO encoded by *cbbL* and *cbbM* genes (Supplementary Figs. 7 and 8) that is crucial for C-fixation[38]. Strikingly, the five most common catalytic metal ions across metal-dependent enzymes with known structure are, by order of importance, Mg, Zn, Fe, Mn, and Ca[39]. They are also the elements the most importantly released by the basalt and supplied to the deep microbial community following $CO_2$ acidification (Supplementary Table 3).

Mantel test and canonical correlations confirmed ions released by basalt dissolution stimulated the growth of autotrophic and heterotrophic bacteria in the aquifer (Fig. 5; Supplementary Table 5). Both trophic types have the potential to impact the fate of the injected $CO_2$. $CO_2$ was autotrophically fixed to biomass solely through the CBB cycle, the prevalent pathway of C-fixation[19]. In addition to autotrophy, $CO_2$ can also be used as electron acceptor to sustain the intermediate steps in microbial transformation of hydrocarbons[40]. Moreover, species from the *Desulfotomaculum* genus blooming in May 2012 assimilate Acetyl-CoA for biomass production through reductive carboxylation (hence using $CO_2$). High ratios of $CO_2$ was already shown to be fixed via this carboxylation reaction[41].

Nonetheless, it is impossible to assess the proportion of $CO_2$ that was assimilated through autotrophic and heterotrophic pathways in the aquifer. First, autotrophic and heterotrophic rates of $CO_2$-fixation may have been highly variable temporally and spatially as highlighted by the fast changes in microbial diversity. Second, the microbial activity was likely not restricted to HN-04 and the aquifer community may also have reacted upstream within the major $CO_2$ plume. Third, ultrasmall cells may be found in groundwater that may at least partially pass through the 0.22 μm filters used for cell capture[42]. In addition, as in the subsurface, most of the biomass exists as attached to the rocks[43], our approach based on groundwater sampling only provides a partial view of the subsurface microbiome and of its reactivity. Accordingly, the impact of the microbial community could have been more important as suggested by (1) the biofilm with high cellular density observed in fractures of the core drilled between the injection well HN-02 and HN-04 in October 2014 (Supplementary Fig. 15), (2) episodic clogging in well HN-02 starting mid July 2012. The drop of injection-well transmissivity was at least partly attributable to the blocking of pores induced by local iron-sulfide precipitation and associated bacterial bloom of *T. denitrificans* that developed as biofilm[44,45]. As iron biofouling is also a widespread and well-recognized problem in aquifers, the bloom of iron-oxidizing *Sideroxydans* sp. could have been also responsible for iron build up in wells and aquifer[46]. Finally, the

sources of organic carbon are not well constrained. Na-flu is a biodegradable heterocyclic compound[47] and could have contributed to the growth of degraders of aromatic compounds. However, the ×20 increase in Na-flu concentrations may not explain the ×500 increase in biomass observed in May 2012. In addition, the degradation of Na-fluorescein would not generate the molecular diversity observed by FT-ICR-MS (Supplementary Fig. 4). Therefore, we postulate that additional sources of organic carbon would have been available in the aquifer. No surface input of soil-derived organic carbon (as highly aromatic and oxidized humic acids) can be considered because the van Krevelen distribution of the organic matter (Supplementary Fig. 4) shows rather aliphatic-type structures[48]. The geothermal gas mixture could not have contributed an organic component related to our findings before July 2012, because prior to then, only pure, commercially-sourced $CO_2$ was used as an injectant and the genes involved with aromatic compound degradation were observed as early as March 2012. Wells were drilled using groundwater from the aquifer, so no organic drilling mud was introduced to the system. Any contribution from drilling fluids corresponds to endogenic organic compounds coming from the basaltic aquifer itself. In the aquifer, organic compounds with a geothermal or volcanic origin would include methane, which can be produced either by chemosynthesis or magma degassing[49]. Abiotic methane as high as 15.2 ppm is observed in the geothermal steam used at the Hellisheidi powerplant (i.e., 0.2% of the emitted gas composition) attesting to the effectiveness of abiotic chemosynthesis of organics in the Hengill volcanic system where $H_2$ is also abundantly detected (up to 68.5 ppm; i.e., 12.3% of the gas composition emitted by the powerplant)[50,51]. Accordingly, chemosynthesis may account for the formation of heavier organic compounds including polycyclic aromatic hydrocarbons (PAHs). Study of volcanic rocks from the Reykjanes peninsula revealed the presence of naphthalene and phenanthrene as well as biphenyl and fluorene in basaltic lava and hyaloclastite[52]. For hydrothermally-altered volcanic deposits, more condensed molecules occurred such as pyrene, benzo(a)pyrene, benzo(ghi) perylene, perylene, reaching concentrations up to 2500 ppb (i.e., two order of magnitude higher than in fresh rock)[52]. This agrees with thermodynamic assessments showing that in volcanic gases condensed *n*-alkanes and PAHs can form metastably from $CO_2$, CO and $H_2$ below ~ 250 °C (i.e., during the lava cooling)[53]. Basalt-trapped PAHs, like metals, may have been released from the host-rock into groundwater following the injection of acidic $CO_2$-charged waters prompting dissolution of the basalt. The released PAHs may have then fed aromatic compound degraders. This hypothesis is supported by statistical analysis showing correlations with DIC, metal and DOC concentrations (Supplementary Table 5). It also agrees with the FT-ICR-MS results that show in February, May and July 2012, in addition to the characteristic complex biological signature, a large number of CHO fragments characteristic of polycyclic aromatics and polyphenol (Supplementary Fig. 4). This molecular signature is indicative of a progressive biological oxidization and hydrogenation of aliphatic and polyaromatic abiotic compounds forming more polar and oxygenated compounds[54]. Overall, this suggests that the aquifer microbial community is used to PAHs as a carbon source. The possibility that PAHs support ecosystems was already raised for gabbro-hosted community along the Mid-Atlantic Ridge, based on the dominance in the gene survey of markers of organic contaminant degradation (up to 45% of the functional genes)[55]. This might then be a general characteristic of microbial communities inhabiting mafic environments.

Overall, as summarized in Fig. 7, the rapid bacterial response to pure-$CO_2$ injection was ruled by several factors that promoted bacterial growth (i.e., basalt dissolution and associated ions'

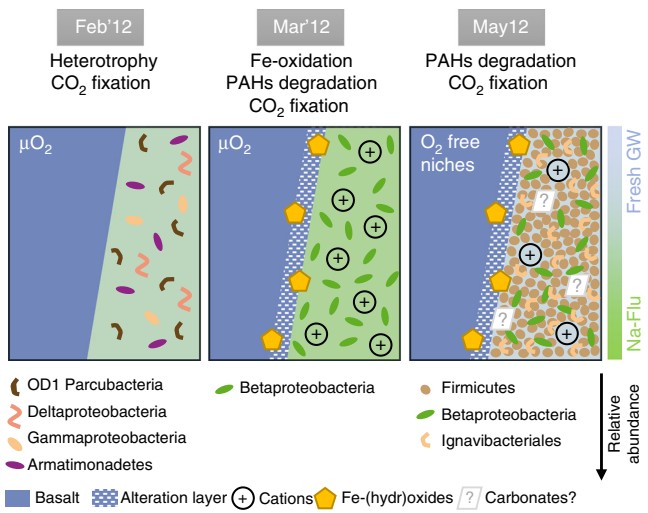

**Fig. 7** Summary of the impact of the pure $CO_2$ injection on the bacterial community hosted at the basaltic Carbfix1 CCS site. The initial population in the groundwater sampled in monitoring well HN-04 was still flourishing in February 2012 (Feb'12) and was mainly composed of heterotrophs living under aerophilic to microaerophilic conditions. In March 2012 (Mar'12), groundwater acidification induced by arrival of the fast-flowing fraction of the pure-$CO_2$ injectant severely reduced bacterial richness, provoked the dissolution of the host-basalt and the release of mineral-forming cations and likely polyaromatic hydrocarbons (PAHs). $Fe^{2+}$ and other ions (e.g., Ca, Mg, Zn) became bioavailable to the ecosystem, as a result favoring the development of Betaproteobacteria including iron-oxidizing autotrophic species related to Gallionellaceae. Because of the high insolubility of ferric iron, iron oxy(hydr)oxides as the byproducts of betaproteobacterial iron-oxidization likely precipitated at the surface of altered basaltic minerals, forming a potentially passivating layer that may have had a deleterious effect on water-rock interaction efficiency[57]. In May 2012 (May12), under more anaerobic conditions, Firmicutes then bloomed along with the *Thiobacillus* species. The well-recognized carbonatogen potential of Firmicutes may have contributed to $CO_2$ mineralization in the form of carbonates at that time[10]. However, mobilization of Fe, Ca, Mg by microbes may have reduced the overall carbonation rates (GW stands for groundwater. Only phyla with relative abundance > 5% according to the number of retrieved pyrosequences are represented)

release, inorganic and organic carbon availability). The stimulated microbial metabolisms can be correlated to changing geochemical conditions driven by host-rock dissolution. The stimulated activities have broad scale implications on the aquifer chemistry, potentially impacting secondary mineralization processes including carbonation. The metabolic activities of some of the inventoried microorganisms, e.g., sulfate-reducing *Desulfotomaculum* species, are alkalinizing and have the potential to enhance carbonation rate[10]. However, due to the involvement of $Fe^{2+}$, $Mg^{2+}$ and $Ca^{2+}$ in their metabolism, bacteria may have immobilized some cations, limiting the precipitation of Ca-Mg-Fe-bearing carbonates. The oxidation of iron by Gallionellaceae and *Thiobacillus* species could have prevented some iron in its divalent form from being incorporated into solid carbonates, promoting instead the precipitation of Fe-oxides/oxyhydroxides and clays, as supported by preliminary observations of the core fractures shown in Supplementary Fig. 15b, c. This secondary coating may passivate mineral surfaces and limit fluid-rock interactions, as does the adsorption of $Fe^{3+}$ ions on mineral surfaces[56]. Additionally, it was shown that the porosity of the silica layer that forms during silicate dissolution strongly depends on iron redox state[57]. By regulating iron speciation up to the local redox state of the aquifer where they develop, ecosystems may have had a

strong impact on basalt dissolution and rates of carbonate formation.

Success of CCS projects relying on mineral storage will depend on the efficiency of fluid-rock interactions governing carbonation. Therefore, it is critical to consider the impact that deep ecosystems and associated biogeochemical reactions have on any habitable rock to ensure long term and safe storage in the form of solid carbonates. $CO_2$ conversion into biomass, as observed here, may offer an alternative pathway for $CO_2$ subsurface entrapment. However, carbon storage as biomass is not desirable as no long-term stability of the biologically stored carbon can be ensured, and controlling the geographic distribution of this biomass along with associated biogeochemical pathways is very unlikely. These statements are valid for both mafic and ultramafic rocks, all being similarly considered as target of prime interest for CCS but also as large microbial habitat[58].

## Methods

**Sampling strategy.** Our monitoring strategy first relied on regular pre-injection sampling from the 9-available monitoring wells[11] during the 4 years that preceded the injections in order to characterize the baseline community. Groundwater sampling was carried out in the deepest wells at key periods during the gas injections (Supplementary Fig. 2). The data set presented here includes six groundwater samples for each of the two selected wells (i.e., HN-01 and HN-04; Supplementary Fig. 1): two before the gas injections (i.e., July 2009 and October 2010), two during the pure $CO_2$ injection (i.e., early February and March 2012), one 2 months after the injection of pure $CO_2$, (i.e., mid-May 2012), and one when the gas-mixture injection had already begun for a few weeks (i.e., July 2012) (Supplementary Table 1 and Supplementary Fig. 2). Well selection was motivated by the following criteria: the monitoring well HN-04 is the closest to the injection well HN-02 and based on tracer experiments, geochemical monitoring and multidimensional reactive-transport modeling[12–14,16], it was the only monitoring well that was affected by the fast-flowing fraction of the $CO_2$ plume. Additionally, HN-01 is a key well for our microbiological monitoring. First, being located 600 m upstream from the injection well, it allowed sampling an aquifer area not affected by the injected gas plume and, as such, can serve as a control well. Second, as part of the CarbFix strategy and in order to promote carbonation of the host-rock, the $CO_2$-bearing gas was mixed and equilibrated at 350 m depth in injection well HN-02 with groundwater pumped down from HN-01 hence leading to instantaneously dissolved-$CO_2$ in the formation waters[2] (Supplementary Fig. 1). Waters and associated microbial community from HN-01 were accordingly continuously injected in the well HN-02 since March 2011 and may have influenced the storage area. The analysis of HN-01 groundwater microbial diversity consequently allowed an investigation of the extent that the HN-01 community impacted that of monitoring well HN-04.

**Microbiological sampling procedure.** To isolate the deep aquifer targeted for $CO_2$ storage from the upper superficial aquifer the CarbFix1 injection site wells are cased (up to 400 and 404 m depth for HN-04 and HN-01, respectively)[11]; an extended description appears in Supplementary Fig. 1. In addition, at 123 (HN-04) and 250 (HN-01) m depth there are submersible pumps that allow groundwater sampling in the targeted basaltic formation and prevent contact with upper formation waters during sampling. Groundwater sampling in monitoring well HN-04 took place after at least 24 h of continuous pumping at ~ 1 l s$^{-1}$ using the down-hole pump. The control well HN-01 was similarly pumped for one hour prior to sampling with a down-hole pump producing water at ~ 70 l s$^{-1}$ (ref. [11]). This long duration and continuous pumping at constant rates produced over 24 h, $8.6 \cdot 10^4$ l and $2.5 \cdot 10^5$ l for HN-04 and HN-01, respectively (i.e., × 8 and × 25 the stagnant water volume for HN-04 and HN-01, respectively). This pre-sample pumping ensured that the samples were representative of fresh groundwater. Pumped volumes are comparable with those considered for similar microbiological sampling carried out at the Wallula CCS pilot site and targeting formation waters of the Columbia River basalts (USA). In that case, $3.3 \cdot 10^4$ l to $6.8 \cdot 10^6$ l were pumped before immediate sample collection[3].

¼" HDPE tubes, previously autoclaved and filled with 5% sodium hypochlorite solution, were connected directly on the wellhead and flushed for 30 min before the addition of sterile 0.22 μm Sterivex™-GP filter units (Millipore, Billerica, MA) containing Express® polyethersulfone membrane. To prevent the filters from clogging or tearing, no more than 12 l of groundwater were filtered per Sterivex™ filter unit, corresponding to duration of 1 to 2 h. At completion, Sterivex™ were filled with sterile absolute ethanol, closed carefully with autoclaved Luer Lock™ plugs and aseptically placed in sterile Falcon™ tubes before being stored at −20 °C until DNA extraction.

In addition to microbiological sampling, temperature, pH and conductivity were measured in the field with handheld pH-meter and conductimeter (compact pH 3310 and Cond 3310 from WTW) providing a precision of ± 0.01 pH unit and

1 μS cm$^{-1}$, respectively. This allowed comparison with the large geochemical monitoring performed in the framework of the CarbFix1 project[11–14] (https://www.or.is/en/projects/carbfix).

**Nucleic acid extraction.** For each sampled well, two filters were manually and aseptically removed from the Sterivex™ under vertical laminar flow hood using sterile forceps, and cut into small pieces with sterile scalpel. Then, total microbial DNA was extracted from filters using UltraClean™ Water DNA Isolation kit (Mo-Bio Laboratories, Carlsbad, CA) following the manufacturer protocol. Total DNA was then stored at −20 °C until amplification.

**Gene quantification.** qPCR amplifications were performed for the 16S-rRNA gene using primers targeting the Bacteria and Archaea domains as a measure of total prokaryotic abundance. The Betaproteobacteria and Crenarchaeota were also quantified by targeting specific regions of the 16S-rRNA gene. The abundance of the ammonia-oxidizing archaea was monitored through the *amoA* gene coding for the α-subunit of the ammonia monooxygenase (Supplementary Fig. 14). All primers were provided by Invitrogen Life Technologies. A description of the primers and the thermal conditions used can be found in Supplementary Table 9. All the reactions were performed in duplicate on a StepOnePlus Real Time PCR system (Applied Biosystems), using the Dynamo Flash SYBR Green qPCR kit (Thermo Fisher Scientific). Fifteen microliter reactions vials contained 1 × SYBR Green Master Mix, 1 μg μl$^{-1}$ bovine serum albumin (BSA), 0.5 μM of each primer and 1 × ROX as a reference dye, and 2 μl of DNA previously concentrated with Vivacon 500 columns (Sartorius). The standard curves were obtained from serial dilutions (from $1 \cdot 10^7$ to $1 \cdot 10^1$ copies/reaction) of linearized plasmids (TOPO TA, Invitrogen Life Technologies) containing standard sequences. The PCR efficiency of the different quantifications ranged between 83.69 and 121.75%. Negative controls resulted in undetectable values in all qPCR reactions. To detect possible inhibitory effects of DNA, $10^5$ copies of plasmid were mixed with sample DNA and quantified with plasmid specific primers T7 and M13r. The obtained cycle thresholds (Ct) values were compared to those obtained when quantifying the plasmid alone by estimating reaction efficiency (i.e., $1-((Ct_{sample+plasmid} − Ct_{plasmid})/Ct_{plasmid}) \times 100$. Reaction efficiency < 95% was considered as indicative of inhibited samples that were then diluted and reprocessed. Conversion factors of 4.1 (16S-rRNA gene copies/cell) were used to convert bacterial gene copies to estimate cell number[59].

**Bacterial and archaeal cloning and Sanger sequencing.** Bacterial 16S-rRNA genes were amplified by PCR using the forward primers B-27F (5′-AGAGTTT-GATCCTGCTCAG-3′) specific for Bacteria with the reverse prokaryotic primer 1492R (5′-GGTTACCTTGTTACGACTT-3′). Archaeal 16S-rRNA genes were amplified by PCR using the forward primers Ar109 (5′-AC(G/T)GCTGCTCAG-TAACACGT-3′), specific for Archaea or ANMEF (5′-GGCTCAGTAA-CACGTGGA-3′), specific for Euryarchaeota, with the reverse primer 1492R. All archaeal OTUs were not constantly retrieved throughout the 5-year-survey period and PCRs were adapted in accordance. In particular, to retrieve Euryarchaeota (OTU_Eury) in July 2009 (i.e., before the gas injections), the forward Archaea-specific primer 21FQ (5′-GGGCGGGCTTCCGGTTGATCCTGCCGGA-3′) was used with the prokaryotic-specific reverse primer 1492R and subsequent semi-nested amplifications were carried out with AMNEF and 1492R as internal forward primer and reverse primer, respectively. All PCRs were performed with a model 2720 or a Veriti® 96-Well thermal cycler (Applied Biosystems [ABI], Foster City, CA). PCRs were carried out using 1 to 5 μl of environmental DNA at concentrations ranging from 4.4 to 25.3 ng μl$^{-1}$ in a reaction buffer volume of 25 μl containing 1.5 mM MgCl$_2$, dNTPs (10 nmol each), 20 pmol of each primer and 1 U GoTaq DNA polymerase (Promega, France). PCRs were performed under the following conditions: 35 cycles (denaturation at 94 °C for 15 s, annealing at 50 °C for 30 s, extension at 72 °C for 2 min) preceded by 2 min denaturation at 94 °C and followed by 5 min extension at 72 °C. Negative controls were carried out systematically for each PCR amplification; all were negative. Amplicons were cloned into TOPO TA cloning kit (Invitrogen Life Technologies) according to the manufacturer instructions. After plating, positive transformants were screened by PCR amplification of inserts using M13r and T7 flanking vector primers. Inserts of the expected size were sequenced by GATC Biotech AG (Konstanz, Germany).

**Bacterial Sanger sequence phylogenetic analyses.** Only high-quality partial sequences (700–800 bp) were retained for subsequent analyses. We discarded sequences of poor quality or potential chimeras. Preliminary distance (neighbor-joining) trees allowed the identification of groups of highly similar sequences (>97% identity) or phylotypes. One representative clone of each phylotype was used for taxonomic affiliation using the SINA software[60]. Partial sequences were compared to those in databases by Basic Local Alignment Search Tool (BLAST)[61].

**Archaeal Sanger sequence phylogenetic analyses.** A total 124 high-quality partial sequences (700–800 bp) were obtained from the 10 16S-rRNA gene libraries and selected for subsequent phylogenetic analyses. Preliminary distance (neighbor-joining) trees allowed for the identification of groups of highly similar sequences (>97% identity) or phylotypes. Several representative clones of the different OTUs were nearly fully sequenced. Complete sequences were assembled using

CodonCode Aligner (www.codoncode.com) prior to phylogenetic analyses. Potential chimeric sequences were identified manually by comparing several portions of the full-length environmental 16S-rRNA gene sequences with sequences of the GenBank database using BLAST[61] in addition to UCHIME[62] and DECI-PHER[63]. Sequences were aligned with the ARB software[64] and then added into the reference tree using the Parsimony tool. Sequences from the clone libraries, along with closely related environmental clones, closest cultivated members and some representative sequences of the major taxa, were selected for phylogenetic tree construction. The resulting sequences were used as input to build phylogenetic trees by maximum likelihood with RAxML[65] using a general time reversible model of sequence evolution, and taking among-site rate variation into account by using a four substitution rate category and an estimated Γ distribution. Base frequency and proportion of invariable sites were empirically estimated. ML bootstrap proportions were inferred using 1,000 replicates.

**Bacterial 454-pyrosequencing.** PCR amplicons of the hypervariable region V1-V3 of bacterial 16S-rRNA gene were generated using the primers 27 F (5′-AGAGTTTGATCCTGGCTCAG-3′) and 534R (5′-ATTACCGCGGCTGCTGG-3′) adapted for 454-pyrosequencing with a 10-bp molecular identifier (MID) tag to identify the samples. In order to minimize the generation of recombinant PCR products, we carried out only 25 PCR cycles. The lower yield was compensated by pooling the products of ten independent PCRs per sample, twice for replicates (Supplementary Fig. 5). All PCRs were performed with a model 2720 or a Veriti® 96-Well thermal cycler (Applied Biosystems [ABI], Foster City, CA). PCRs were carried out using 1 to 5 μl of environmental DNA at concentrations ranging from 4.4 to 25.3 ng μl$^{-1}$ in a reaction buffer volume of 25 μl containing 1.5 mM MgCl$_2$, dNTPs (0.2 mM each), 10 pmol of each primer, and 1 U Platinum Taq DNA polymerase (Invitrogen). PCR reactions were performed under the following conditions: 25 cycles (denaturation at 94 °C for 15 s, annealing at 55 °C for 30 s, extension at 72 °C for 2 min) preceded by 2 min denaturation at 94 °C, and followed by 5 min extension at 72 °C. Negative controls were carried out systematically for each PCR amplification experiment; all were negative. For each sample, the ten pooled PCR were purified and concentrated in 30 μl using a UltraClean® Microbial DNA Isolation Kit (Mo-Bio Laboratories, Carlsbad, CA). DNA concentration was measured through DNA absorbance at 260 nm in a ultraviolet–visible spectrophotometer (NanoDrop 2000, Thermoscientific) before pooling samples equimolarly. Pyrosequencing (Beckman Coulter Genomics, Takeley, UK) was performed using the Roche GS FLX platform (454 Life Sciences, Branford, CT) with the Titanium LIB-A kit for bi-directional amplicons sequencing. Supplementary Figure 5 shows that results obtained for the two replicates are self-consistent.

**Pyrosequence analysis.** Pyrosequence data were processed using the QIIME pipeline[66] following standard practice and algorithms incorporated into QIIME. Reads were quality filtered by removing reads with unresolved bases and/or anomalous read length. Sample barcode and primer sequences were trimmed from the proximal end of the sequences without any truncation of the sequences on the 3′ end, as recommended for proper use of Denoiser[67]. Reads shorter than 300 nucleotides or containing one or more ambiguities were removed. This process was done twice in order to apply Denoiser on both the forward and the reverse set of sequences. Reverse primer, barcode and the following sequence were then removed on both sets. Reverse sequences were reversed before concatenating the two sets. OTUs were picked using Uclust[68]. Alignment was performed with PyNAST[69] with the SILVA 111 reference database[70] before chimera checking using ChimeraSlayer[71]. Sequences were assigned taxonomy at 97% similarity using the RDP classifier 2.2[72] and the SILVA 111 reference database, with a confidence >50%. Alignment was then filtered with an entropy threshold of 10% and positions with more than 80% gap were removed. Tree was built with FastTree 2.1.3[73] for subsequent analyses. Singletons and sequences non-affiliated to an OTU were removed.

Unifrac[74] was used for Principal Coordinates Analysis (PCoA) following standard practice adapted to the samples, with equalized sampling depth from rarefaction analyses removing sample heterogeneity. Beta-diversity analysis was performed using jackknife replicates in order to estimate the uncertainty in PCoA plots and hierarchical clustering of microbial communities. Beta significance test was done with 1,000 permutations on the overall data set and for each well separately, and indicated that the weighted beta-diversity metrics were not significant ($p$-value » 0.01 for each pair of environment—data not shown). In accordance, unweighted beta-diversity metrics were only considered here.

Canonical correlation analysis was performed to investigate relationships between OTU relative abundances and environmental parameters in both control well HN-01 and monitoring well HN-04. PERMANOVA tests were previously applied on each variables with adonis function of R-vegan package[75]. It allowed selecting significantly correlated parameters (i.e., $p$-value < 0.05). Correlations between the qualitative Unifrac distance matrix[74] and the corresponding geochemical metadata were explored using Mantel test with 9,999 permutations, based on Pearson's product moment. Pearson correlation coefficients (r-values) were evaluated to test linear correlations between the geochemical data and the abundance of taxonomic groups. All statistical analyses were performed using a paired-end reads OTU table normalized to the lowest sampling depth and

computed on R-Studio v.3.3.2[76] using vegan v.2.4-2[75], and ggplot2 v.2.2.1[77] packages.

**PCR-cloning of functional genes**. Among the series of functional genes coding for inorganic carbon assimilation and organic carbon degradation that were studied by selected amplifications (listed in the caption of Supplementary Table 4), only three led to successful amplifications: first, *cbbL* and *cbbM*, respectively encoding form I and form II ribulose-1,5-bisphosphate carboxylase/oxygenase (RuBisCO), a key enzyme for autotrophic $CO_2$ fixation; second, with two different sets of primers (pheU and PHE), the genes coding for the largest subunit of multicomponent phenol hydroxylases (LmPHs) involved in the degradation of phenolic compounds. PCR primers and thermal conditions are described in Supplementary Table 4. Negative controls were carried out systematically for each PCR amplification; all were negative. Amplicons were cloned into TOPO TA cloning kit (Invitrogen Life Technologies) according to the manufacturer instructions. After plating, positive transformants were screened by PCR amplification of inserts using M13r and T7 flanking vector primers. Inserts of the expected size were sequenced by GATC Biotech AG (Konstanz, Germany). A total of 21, 2, 7, and 6 high-quality partial sequences were, respectively, obtained for *cbbL*, *cbbM*, pheU, and PHE primers and retained for subsequent phylogenetic analyses. Potential chimeric sequences were identified manually by comparing several portions of the full-length environmental gene sequences with sequences of the GenBank database using BLAST[61].

Sequences from the clone libraries, along with closely related environmental clones, closest cultivated members and some representative sequences of the major taxa, were aligned and selected for phylogenetic tree construction using the Maximum Likelihood method using the MEGA software[78].

**Metagenomic library preparation and sequencing**. Metagenomic sequencing was done by the Josephine Bay Paul Center for Comparative Molecular Biology and Evolution at the Marine Biological Laboratory (Massachusetts, USA). The concentrations of the two genomic DNA samples (HN4_march12, HN4_may12) were determined by Quant-iT Picogreen dsDNA assay (Life Technologies, Carlsbad, CA). Samples were fragmented to ~ 170 bp using a Covaris S220 Focused-ultrasonicator (Covaris Inc. Woburn, MA) and metagenomic libraries were prepared according to the Nugen Ovation® Ultralow Library system protocol (NuGen Technologies, Inc. San Carlos, CA). Prior to sequencing, metagenomic libraries were visualized on an Agilent DNA 1000 Bioanalyzer chip (Agilent Technologies, Santa Clara, CA) and quantified using a KAPA SYBR® FAST Universal qPCR Kit (KAPA Biosystems, Boston, MA). Paired-end sequencing (2 × 113 bp) was performed on an Illumina HiSeq 1000 (Illumina, Inc. San Diego, CA). Base calls, sample demultiplexing, quality scores, and individual FASTQ files for each sample were generated on a CASAVA 1.7 + pipeline (Illumina Inc. San Diego, CA).

**Metagenomic sequence assembly and analysis**. Forward and reverse reads from paired-end sequenced DNA libraries were assembled using the FLASH software[79] with default parameters (minimum overlap, 10 nt; maximum allowed ratio between the number of mismatched base pairs and the overlap length, 0.25). Merged paired-end reads where assembled without further filtering with MEGAHIT v0.3.3-a software[80] using 27 to 123 k-mers range in steps of 10, all other parameters set to their default values. Gene prediction on subsequently generated contigs was performed with Prodigal version 2.60, metagenomic mode[81]. The sequence pool thus generated was screened with RefSeq nr (release 68, Nov 3, 2014) and COG database updated in 2014[26] using USEARCH v6.0.307 software[68] and –ublast/-evalue < = 1e-04 parameters. Normalization of bacterial metagenomic hit counts was performed by comparing with a set of 40 well known single-copy gene families universally distributed among prokaryotic genomes, which are present in the COG database[82]. Determination of *cyc2* gene homologs differential expression was performed by direct sequence similarity search of 2 *cyc2* homologs namely AKN78226.1 (Cyc2PV-1 from *Mariprofundus ferrooxydans*) and ADE10507 (Slit_0265 *Sideroxydans lithotrophicus*), against the appropriate full reads data sets.

**Inductively coupled plasma quadrupole mass spectrometry**. Elemental concentrations of filtered and acidified groundwater were characterized at low resolution using an Agilent 7900 ICP-QMS in pulse counting mode. Sample introduction was achieved with a micro nebulizer (Micro Mist, 0.2 ml min$^{-1}$) through a Scott spray chamber. All elements were measured using a collision reaction interface with helium gas (5 ml min$^{-1}$) to remove isobaric interferences.

**Fourier transform ion cyclotron resonance mass spectrometry**. In order to assess the molecular diversity of organic compounds present in groundwater through time, non-targeted ultrahigh-resolution molecular analysis of the solvent-accessible organic fraction were performed. Spectra were acquired in negative ionization mode with a Bruker SolariX FT-ICR-MS equipped with a 12 T superconducting magnet and coupled to an Apollo II electrospray ionization source. For this purpose 200 ml of 0.2 μm filtered samples were prepared by solid phase extraction (SPE)[83]. All samples were acidified with hydrochloric acid (32%, p.a., Merck KGaA, Darmstadt, Germany) to pH 2 and passed through SPE cartridges (Bond Elut PPL 1 g, Agilent Technologies, Waldbronn, Germany). Afterwards cartridges were rinsed with acidified purified water (MilliQ-Integral, Merck KGaA,

Darmstadt, Germany) and dried under vacuum. Samples were eluted with 3 ml methanol (Chromasolv® LC-MS grade, Sigma Aldrich, Taufkirchen, Germany). Methanolic extracts were diluted 1:50 with methanol and continuously infused with a flow rate of 120 μl h$^{-1}$. The spectra accumulate 300 scans with 4 M data points in the mass range from *m/z* 147 until 1000. Spectra were calibrated internally in the presence of natural organic matter, resulting in a mass accuracy better than 0.1 ppm.

**Data availability**. The authors declare that the data supporting the findings of this study are available within the article and its Supplementary Information File. Archaea Sanger sequencing data are deposited at the GenBank of the National Centre for Biotechnology Information under accession numbers KJ867248 to KJ867371. Bacterial Sanger sequences received accession numbers from KU685482 to KU685504 and KX276716 to KX276768. Sequences from Bacteria 454-pyrosequencing and both metagenomic studies are available at the Sequence Read Archive (SRP042183, SRR3731039, and SRR3731040, respectively). Sequences reported in this paper for *cbbL*, *cbbM* genes and for genes coding for the largest subunit of multicomponent phenol hydroxylases (LmPHs), respectively received the following GenBank accession numbers: KX290744 to KX290758, KX290759 to KX290760 and KX290761 to KX290773.

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

## Acknowledgements

We are grateful to the CarbFix partners (https://www.or.is/english/carbfix-project) for the opportunity to implement this microbiological survey via access to the Hellisheidi $CO_2$ injection pilot. We gratefully acknowledge the financial support of TOTAL, SCHLUMBERGER and the French ADEME agency along with NSF and DOE for the drilling (DOE Award Number: DE-FE0004847-PI: J. Matter, Earth Institute/Lamont). We warmly thank P. Lopez-García, D. Moreira, P. Deschamps, and C. Bachy for helpful discussions and guidance on data handling and analysis along with R.L. Moore. We thank E. Örn Þrastarson for assistance during groundwater sampling, M.C. Marinozzi, C. Lemonnier, P. Henri, and A. Michel for assistance during experiments, along with P. Bénézeth for support throughout this work. Our thanks also go to scientists from SCHLUMBERGER and TOTAL for constructive comments on the manuscript. The research leading to these results has also received funding from the French national agency ANR through the CO2FIX (ANR-08-PCO2-003-03) and the deepOASES (ANR-14-CE01-0008-01) projects, the People Programme (Marie Sklodowska-Curie Actions) of the European Union's Seventh Framework Programme FP7/2007-2013/ under REA grant agreement no. 624382 and the Deep Life Community of the Deep Carbon Observatory funded by the Alfred P. Sloan Foundation. Metagenomic sequencing was performed thanks to a DCO Census of Deep Life project. This is IPGP contribution no. 3880.

## Author contributions

B.M. and E.G. designed the project and organized its implementation. E.G., B.M., P.L., R.T. participated to the field work. L.L., P.L., E.G., R.T., B.M., P.S.K., J.U. performed the experiments. Y.Z. performed metagenomic sequence assembly and analysis. A.L. contributed to the statistical analysis. B.M., E.G., P.L., R.T. interpreted the results and wrote the manuscript. S.R.G., J.M.M., M.S., E.H.O., H.A.A., K.G.M., S.O.S., E.S.A., and I.G. provided access to the CARBFIX infrastructure along with background on geochemical data.
