## [Peer Review File · Nature Communications]

Reviewers' Comments:

Reviewer #1 (Remarks to the Author)

This paper presents the first microbiological analysis of the outcome of a large scale experiment to test the capacity of subterranean basaltic rocks to be used as a reservoir for the long-term storage of CO₂. This work required substantial coordination between this team doing the microbial monitoring and the engineers and earth scientists involved in the carbon capture aspect. While the overall results of this test may not have been what was initially expected, as presented here the microbiological findings are entirely consistent with what the actual geochemical conditions were that ended up being created underground. Essentially, the highly concentrated CO₂ caused water-rock reactions with the relatively fresh basalt that lead to a large release of Fe(II), because this is a system with oxic groundwater, a large bloom of oxygen-dependent autotrophic Fe-oxidizing bacteria ensued that caused some blockage of the pore space in the rock, but that also likely consumed the O₂ leading to a secondary bloom of more anaerobic microbes including sulfate reducers that took advantage of organic matter, the source of which seems to be a bit of mystery, and resulted in a substantial increase in subterranean biomass. This team used all the requisite geochemical and cultivation-independent microbial tools to provide monitoring that quite clearly shows the microbial outcome of the experiment.

The work is important, novel, at least in terms of the application, and integrates both the geochemistry, hydrology, and microbiology into a coherent story. That said the paper suffers substantially from a confusing style of writing that makes interpretation of the findings difficult to follow. Some examples, but by no means all the instances, are given below. The paper would benefit greatly from having a skilled editor revise it.

General Comments.

The opening paragraph could briefly mention the alternative ways of doing subsurface CCS, and then the 2nd paragraph could be tightened up to only discuss how this particular method is supposed to work. In the introductory paragraphs, there should also be some general explanation of why this site was chosen for this CCS test, was it because the powerplant was close by, or because the rock type was interesting, or a combination of the two.

The enhancement of genes involved in organic matter degradation is interesting, especially aromatics. They seem to imply this may have been a response of the community to the tracers, however there is no real accounting of the potential concentration of electron donors, i.e. how much organic matter may have been present, e.g. from the tracers. Furthermore, they do not discuss what other sources of organic matter might have been present, one likely source is the flue gas mixture that may have contained aromatic or other hydrocarbons, apparently it was 'cleaned up' before injection, but how efficient was this?

Data availability, The MS is unacceptable until all molecular data is deposited in the appropriate publicly available data repositories.

Specific comments.

l42, change 'could lock up' to 'have the potential to...'

l52, presumably you mean the microbial communities living in these basalt-systems.

l57, This paragraph should really start the results section, instead of being in the introduction, Fig 7 could be used here.

I59, Why this particular combination of gases?

L94, Here and elsewhere, what is meant by 'fast-flowing' please provide a velocity or range of estimated velocities.

I142-145. As written, the meaning of this sentence is difficult to follow

I145, 'constantly retrieved', do you mean present in low abundance?

I148, more resistant? Or was a subset selected for growth by the changed conditions?

I162, Here and elsewhere, according to Linnean principles, only genus and species are italicized.

Paragraph starting with I153. Please see the paper of Barco et al, AEM 2015 regarding the *cyc2* gene, which is present in Sideroxydans and for which there is evidence of it being involved in Fe-oxidation. The authors need to specifically determine if this *cyc2* homolog from Sideroxydans is present in their dataset, also see the recent paper by Beller et al ISME J 2016, where evidence for Gallionellales was found in an aquifer, and a *cyc2* homolog was abundantly expressed.

I 170, acidothiobacilli is an anachronistic term, better to say strains related to Acidithiobacillus.

I174, It's not clear what the point is here, the conversion of DIC to biomass must be very small relative to the amount of CO₂ that is being introduced into the system. A better accounting or approximation of how much biomass may have actually resulted from this process would be useful.

I221-226. The writing is very awkward here, so it's difficult to follow the point they are making.

I230-231, This is really a discussion point.

I231, conversely, poor word choice, either, At the same time, or Concurrently, would be better.

I238-240, It's not clear what is meant here, T.d and S.I share little overall gene homology, so either their analysis is incorrect, or they are discussing a specific gene or genes where these organisms may show more homology.

pg13, I'm missing the importance of Fig 5 here that shows most genes that except for acetyl Co-A genes, are expressed in the same range or more highly expressed in March compared to May, suggesting the system was already primed for the biomass bloom that they saw in May. This does not seem to be consistent with their explanation of a 'bloom' in May, especially of aromatic degradation.

I253-256 Protein motif is incorrect, this work is based on metagenomics not proteomics.

I256, what is an aromatic cycle?

I262-263, More unspecific text, what's meant by 'some of the importantly'

I264-265, It's not clear what is meant by 'close arrival of the fast-flowing fraction...'

I288, resistance would be a difficult thing to prove in this case, better to say adapted or more adaptable

1st paragraph of discussion. It's curious they begin the discussion here. This argument for micro-

nutrient availability only works if the system is limited for micro-nutrients rather than electron donor or acceptor. In general rock-hosted systems are not thought to be limited for mineral-based micronutrients, except perhaps N; they provide no references to support their case. Because they do not provide any information on actual concentrations of either electron donors or acceptors in the system the argument that this is a micro-nutrient limited system is very weak and should be re-considered.

l315 - 317. Should say something about the fluorescein concentrations and dilution, is it reasonable to assume there was enough organic carbon in the fluorescein to support significant growth?

l331-333, clogging discussion, it seems another likely clogging culprit was the bloom of Sideroxydans, FeOB are common problems in water aquifers due to clogging.

l384-386, meaning of sentence is unclear as written.

l390, please provide depth values.

l506-507, why not just briefly outline the quality control process rather than saying it was 'rigorous'?

l590. Data availability, The MS is unacceptable until all molecular data is deposited in the appropriate publically available data repositories.

Fig 6 legend and elsewhere, it's unclear what is meant by inhibition, since they did not track individual populations how is it know whether this is inhibition, or simply outgrowth of specific populations.

Figure 7, It seems like this figure should be figure 1, since it sets the physical setting. However, it might best as supplemental. I cannot make out what meant in 7e and the explanation does not help. The Fe-oxide slimes are interesting, FeOB are known for producing unique morphological structures, they should state whether or not any of these were observed by microscopy.

Reviewer #2 (Remarks to the Author)

The language is generally good but there are a few stylist points that could be improved. I have noted some of these in the minor comments but the manuscript would benefit from proofreading.

I realize that much of the information is published, but the text from line 57 to 81 feels like methods. Could this be edited to be more of an introduction?

In general, the manuscript would be improved throughout by replication of data and addition of statistics and significant correlations between the tested parameters. The first example is the need to add standard deviations data from Table S2 to Fig. 1 and the text in lines 96-100 and then state where the differences were significant. In the same manner, please present statistics for ion release (line 102) and statistically significant differences presented with number of replicates, SDs or confidence limits.

The 16S rRNA gene community profile would also benefit from replication and statistics. This would show if the differences between HN-01 and HN-04 were significant.

Why were genes involved in the CBB cycle targeted for carbon dioxide fixation (lines 135)? This is the most "energy expensive" method for carbon fixation and genes for other carbon fixation processes should also be investigated. In fact, the authors later discuss the reductive carboxylate

cycle and this should also be included.

Has HPIP been detected in other iron oxidizing populations than the acidithiobacilli (line 169)? Otherwise this would be a surprising finding as species in this genus are obligate acidophiles.

Could the source of aromatic carbons be from e.g. humic acids from the photosynthetically fed surface, from the Na-fluorescein (line 198), or drill fluids? Is it known how long the water in the present system has been partitioned from the surface and which if any of these possibilities is the most likely?

Why were the metagenomes not binned to "metagenome assembled genomes"? This would have allowed metabolic functions suggested by the identified genes to be ascribed to phylogenetic taxa. It would also alleviate the point below.

In general, please be careful in assigning metabolic functions to populations based on 16S rRNA gene alignments. There are many cases where high 16S gene similarities do not correspond to similar functions. The results and discussion should be edited to reflect this limitation.

Lines 393-400: What was the volume of water needed to be pumped to ensure that the stagnant well water was removed from HN-04 and HN-01? Based on these numbers, how many volumes were removed by pumping 8.6×10^4 litres and 2.5×10^5 litres from HN-04 and HN-01? The actual amount pumped is not the critical point; it is that the pumped volume should be at least three times the stagnant well water volume.

Lines 401-403: I realize that the samples were taken several years ago, but recent studies have shown ultrasmall cells in oligotrophic groundwaters that will (at least partially) pass through a 0.22 μm filter (see: Luef et al (2015) Nat Commun 6:doi: 10.1038/ncomms7372). It would be interesting to test for small cells and this could be commented upon.

Lines 420-422: Did the authors test for amplification with archaeal 16S primers? If not, this should be noted and the text altered to reflect the qPCR abundance estimates reflected the total bacterial and not prokaryotic abundance.

Line 517: Why were "sequences non-affiliated to a phylum" removed? Would this not have removed any novel candidate phyla identified in the samples?

Minor comments

Line 27: I suggest: "In this study, we have carried out the first..."

Line 28: I am not sure what a "high" response is. Please give a quantitative measurement?

Line 31: "... resulted in a marked decrease in microbial richness."

Line 37: This is a long sentence with many clauses. Please split into two to simplify the language.

Line 37: "...would reconcile the use..."

Line 40: Once again, the sentence has many clauses and could be simplified. I will not comment on this again but please read through the manuscript and edit further instances.

Line 55: Replace "retroactively" with "subsequently"?

Line 75: "...study focuses on..."

Line 76: "... which a few percent..."

I will not make language style suggestions after this point, but feel the text could be proofread.

Line 130: Was the pH drop significant? Please support the text with data.

Line 143: Please avoid adjectives like "drastic". Instead, please provide numbers and statistics to support your statements.

Line 148: I would not say that Betaproteobacteria "appeared more resistant to the pH decrease and the rise in metal concentrations" but rather these conditions allowed these populations to proliferate.

Line 165: "cytochrome"

Line 167: The c in "c-type" should be in italics.

Line 224: I would suggest the increase in cell numbers was "likely" due to the carbon dioxide injection.

Line 273: The data suggest that the conditions did not favor autotrophic species that fixed carbon via the CBB cycle. Other carbon fixation systems should also be investigated.

Line 278: What were the potential roles for the two detected archaeal populations?

Line 289: This is a similar statement as made in line 148.

Line 292: Ferrous iron likely constituted an electron donor for the populations. This has not been proven.

Line 295: Why was zinc and magnesium release only beneficial for heterotrophs?

Line 307: Basalt dissolution was not involved in the general bloom of bacteria. Instead, I would say that a bloom may have occurred as a consequence of the ions released by the dissolution.

Line 310: How can CO₂ fixation be linked to heterotrophy? If the Desulfotomaculum genus is using the reductive carboxylate cycle then it is an autotroph. Also, please see line 322 where it is stated "heterotrophic rates of CO₂ fixation".

Line 390: Please define "at great depth".

Line 391: What do the authors mean by "reservoir conditions"?

Line 408: Is there any evidence that loss of groundwater pressure would result in precipitation of dissolved ions, altering both pH and redox potential?

Line 410: I do not think "confrontation" is the correct word. Please edit.

Line 433: How were the obtained cycle thresholds (Ct) defined as not significantly different to those obtained when quantifying the plasmid alone? Please give details.

Lines 450 and 492: Give the DNA concentrations (and not volume).

Line 576: State what "Electrospray ionization Fourier transform ion cyclotron resonance/mass spectrometry" was used to detect.

Line 656: The "2" should be in subscript.

Line 666: The reference title should not be capitalized.

Line 782: Please fix "This is IPGP contribution n° XXX".

Fig. 3: Are the size of the circles representative of up to the given percentage i.e. the smallest circle represents 1 to 5% while the second smallest is 5 to 10%? Please give more details.

Fig. 5: Are these not genes and therefore, should be in italics?

“High reactivity of deep biota under anthropogenic CO₂ injection into basalt”

Responses to the Referees

We would like to thank the Referees for careful reading of the manuscript and their constructive comments. Changes and additions were made accordingly in the revised version of the manuscript. Responses to the Referees' comments are provided below in blue and in italics. The line numbers generally refer to the revised version of the manuscript. Otherwise, it is stipulated.

Reviewer #1 (Remarks to the Author):

This paper presents the first microbiological analysis of the outcome of a large scale experiment to test the capacity of subterranean basaltic rocks to be used as a reservoir for the long-term storage of CO₂. This work required substantial coordination between this team doing the microbial monitoring and the engineers and earth scientists involved in the carbon capture aspect. While the overall results of this test may not have been what was initially expected, as presented here the microbiological findings are entirely consistent with what the actual geochemical conditions were that ended up being created underground. Essentially, the highly concentrated CO₂ caused water-rock reactions with the relatively fresh basalt that lead to a large release of Fe(II), because this is a system with oxic groundwater, a large bloom of oxygen-dependent autotrophic Fe-oxidizing bacteria ensued that caused some blockage of the pore space in the rock, but that also likely consumed the O₂ leading to a secondary bloom of more anaerobic microbes including sulfate reducers that took advantage of organic matter, the source of which seems to be a bit of mystery, and resulted in a substantial increase in subterranean biomass. This team used all the requisite geochemical and cultivation-independent microbial tools to provide monitoring that quite clearly shows the microbial outcome of the experiment.

The work is important, novel, at least in terms of the application, and integrates both the geochemistry, hydrology, and microbiology into a coherent story. That said the paper suffers substantially from a confusing style of writing that makes interpretation of the findings difficult to follow. Some examples, but by no means all the instances, are given below. The paper would benefit greatly from having a skilled editor revise it.

We thank Reviewer#1 for his/her comments. The paper was revised by a skilled native English colleague.

General Comments.

The opening paragraph could briefly mention the alternative ways of doing subsurface CCS, and then the 2nd paragraph could be tightened up to only discuss how this particular method is supposed to work. In the introductory paragraphs, there should also be some general explanation of why this site was chosen for this CCS test, was it because the powerplant was close by, or because the rock type was interesting, or a combination of the two.

The opening paragraph does now mention the alternative ways of doing CCS and the description of mineral storage was tightened in accordance (lines 42-51): “CO₂ geological storage at depth in depleted oil reservoirs, saline aquifers or (ultra)mafic rocks is considered as a solution which may reconcile the use of fossil fuels with the control of greenhouse-gas emissions and associated environmental consequences.

The method of CO₂ storage, as supercritical, gaseous, dissolved in formation water or converted into solid carbonates, is dependent on rock lithology, targeted depths and associated pressures and temperatures. In recent years, subsurface storage into basalts or peridotites, rich in calcic and ferromagnesian silicates, has been considered because these rocks have high potential for secure and long-term carbonation¹⁻². Through the dissolution of their silicate components, (ultra)mafic rocks have the potential to trap, as precipitated Ca-Mg-Fe-carbonates, significant quantities of CO₂ in Earth's crust."

The rationale for site selection is also more explicitly explained (lines 65-67): "Our study focuses on the CO₂-injection site, associated with the Hellisheidi geothermal powerplant (SW-Iceland; Fig. S1), developed in the framework of the Carbfix project to assess the feasibility of carbon capture and in situ mineral storage in basalt^{2,11}."

Full information about the site can also be found in the caption of Supplementary Fig. S1.

The enhancement of genes involved in organic matter degradation is interesting, especially aromatics. They seem to imply this may have been a response of the community to the tracers, however there is no real accounting of the potential concentration of electron donors, i.e. how much organic matter may have been present, e.g. from the tracers. Furthermore, they do not discuss what other sources of organic matter might have been present, one likely source is the flue gas mixture that may have contained aromatic or other hydrocarbons, apparently it was 'cleaned up' before injection, but how efficient was this?

We thank Reviewer#1 for her/his suggestion. We fully agree that the source of the organic matter sustaining the enhancement of genes involved in the degradation of aromatic compounds is a critical point. We drafted a discussion on its likely origins in the former version of the manuscript while acknowledging however that these sources were not well constrained and difficult to identify (former lines 313 to 323). We now elaborate on this point (from lines 362 to 404) by further discussing the FT-ICR-MS data shown in Supplementary Figs. S3 and S4 and by introducing insights from literature.

*As discussed in the previous version of the manuscript, the first evident and likely source is indeed the **Na-fluorescein** (Na-flu), a biodegradable heterocyclic compound. To better compare its increase in HN-04 groundwater with the amplitude of the bacterial bloom, Na-flu concentrations are now more clearly provided (lines 115-117, 237, 297) as a complement to its temporal evolution shown in Supplementary Fig. S2c. Note nonetheless that those could have been underestimated as Na-flu absorbance measurements could have been hampered by groundwater pH values and elevated Fe concentrations, especially in March 2012 where iron showed its highest concentrations (Naim et al., 1986; Sjöback et al., 1995; Doughty, 2010). Anyhow, while bacteria may have benefited from this source of carbon, the x20 increase in Na-flu concentrations may difficultly explain by itself the x500 increase in biomass observed in May 2012. In addition, the degradation of Na-Fluorescein would not generate the organic molecular diversity established by FT-ICR-MS and shown in Supplementary Fig. S4. Consequently we may postulate that additional sources of organic carbon would have been available in the aquifer during the monitored period.*

Alternatively, as also suggested by Reviewer#2, we then considered the following sources as potential candidates:

- **humic acids from the photosynthetically fed surface:** the van Krevelen distribution of the organic matter found all along the survey (Supplementary Fig. S4) shows rather aliphatic-type structures not typical of groundwater organic matter/humic acids that are more aromatic and highly oxidized (Einsiedl et al., 2007). Thus, this limits the hypothesis of a photochemically fed surface origin for the organic compounds. Supplementary Fig. S4 was modified accordingly to display the distribution expected for dissolved organic carbon derived from the surface biomass and now clearly shows the

discrepancy;

- **drilling fluids:** injection and monitoring wells were drilled using groundwater from the aquifer as drilling fluids, thus implying that no organic muds was added to the system. If a contribution is expected from drilling fluids, it then corresponds to endogenic organic compounds coming from the deep basaltic aquifer itself (please see below);
- **flue gas mixture:** a contribution of the geothermal gas mixture can only be considered for the groundwater community sampled in July 2012 because otherwise, pure commercial CO₂ was used. As genes involved in organic compound degradation (especially aromatics) are detected as early as March 2012, the gas mixture harnessed by the power plant did not contribute to the observed microbial blooms. We cannot exclude a contribution for the latest sampling period (i.e. July 2012) but with the exception of methane, heavier hydrocarbons including aromatic compounds have never been characterized by the Carbfix consortium neither in the gas emitted by the power plant nor in the one collected after the separation process. It hence makes difficult to assess its contribution for July 2012. Their presence in the injected gas mixture would depend on each compound solubility as during the purification process, the pilot gas station separates CO₂ from the geothermal gas coming from the condensers of the power plant by sequential extraction. However as no combustion is performed in the plant, organic impurities, if present, should reflect the ones that can already be found in the pristine geothermal gas emitted by the Hengill volcanic system and circulating sporadically in the aquifer (please see below). To better document the reader on the cleaning procedure, the details provided above are now included in the caption of Supplementary Fig. S1;
- **endogenic organic compounds coming from the rock itself:** although the main constituents released by geothermal and volcanic manifestations are mantle H₂O and CO₂, CH₄ can also occur from some ppmv to a few % units, being produced either by inorganic reactions (i.e., chemosynthesis), magma degassing and/or thermal breakdown of organic compounds in sediments (Etiope et al., 2007). The presence of abiotic methane as high as 15.2 ppm in the geothermal steam used by the Hellisheidi power plant (i.e. 0.2% of the composition of the emitted gas) attests for abiotic chemosynthesis of organic compounds to be an effective process in the Hengill volcanic system where H₂ is also abundantly detected (up to 68.5 ppm; i.e. 12.3% of the gas composition emitted by the power plant) and no sediments were observed (Gunnarsson et al., 2013; Scott et al., 2014). In accordance, chemosynthesis could also account for the formation of heavier organic compounds including polycyclic aromatic hydrocarbons (PAHs). The study of basaltic lava and hyaloclastites from the Reykjanes peninsula (Iceland) revealed the predominance of naphthalene and phenanthrene as well as biphenyl and fluorene. For the hydrothermally altered volcanic deposits, more condensed molecules occur such as pyrene, benzo(a)pyrene, benzo(ghi)perylene, perylene (Geptner et al., 1999a,b, 2005) reaching concentration of up to 2500 ppb (i.e. 2 order of magnitude higher than in the fresh rock). This is in agreement with thermodynamic assessment that showed that in the present Earth's volcanic gases, condensed n-alkanes and PAHs can form metastably from CO₂, CO and H₂ below ~250°C (i.e. during the lava cooling; Zolotov & Shock, 2000). Thus, similarly to what was observed for the mineral-forming metals that have been released into the groundwater following its acidification by CO₂ and the subsequent dissolution of the basalt (Supplementary Table S3 and new Fig. S2), basalt-trapped PAHs could have been released the same way, hence explaining why genes coding for enzymes involved in aromatic compound degradation are the most abundant in the metagenomic data. This hypothesis is supported by the similar evolutions of DOC, Mg, Ca and Fe concentrations in HN-04 groundwater after the gas mixture injection (new data added to Supplementary Fig. S2) and new statistical analysis (including Mantel tests) showing correlations between DIC, metal and DOC concentrations (new Supplementary Table S5). This is also in agreement with the FT-ICR-MS results that show during the gas injections, in February, May and July'12, in addition to the characteristic complex biological signature, a large number of CHO fragments characteristic of polycyclic aromatics

and polyphenols (Supplementary Fig. S4). Similarly to what was described by Rainer et al. (2014) and Dvorski et al. (2016), such a molecular signature provides clues for a progressive biological oxidization and hydrogenation of aliphatic and polyaromatic compounds likely released by the basalt, hence forming more polar and oxygenated compounds. Although CxHy compounds are not ionized by using electrospray, their hydroxylated and carboxylated degradation products are in that respect well highlighted in the CHO, CHOS, CHON, CHONS molecular series at H/C<1 (Supplementary Fig. S4).

Overall, this suggests that the Hellisheidi microbial community may be used to PAHs as carbon source. No clearly-autotrophic groups of bacteria were detected in the aquifer before the injection although the Dissolved Inorganic Carbon (DIC) concentrations were 50 to 90 times higher than that of the Dissolved Organic Carbon (DOC) in October 2010 and July 2009, respectively (Supplementary Table S2 and new Fig. S2). This might be a characteristic feature of microbial communities inhabiting mafic environments and subjected to volcanic activity. The possibility that aromatic hydrocarbons support ecosystems in similar contexts was already raised for gabbro-hosted community along the Mid Atlantic Ridge (so, far from any anthropogenic influence), based on the dominance in the gene survey performed by Mason et al. (2010) of markers of organic contaminant degradation (up to 45% of the functional genes). The capabilities of microbes to rely on such compounds also closely relate to the ability of the basalt to provide metallic cofactor (e.g., Zn and Fe) that are mandatory for the enzymes involved in aromatic hydrocarbon degradation.

In order to better discuss the likely sources of organic compounds, we modified the manuscript accordingly along with Supplementary Figs. S2 and S4 and associated captions.

Concerning the quantification of the process efficiency, this last is however hampered at this stage by several limitations listed below. It will need, among others, analogical experiments aiming at assessing how much organic matter can be released from the rock and can be used to sustain heterotrophs (purpose of work currently in progress) along with drilled core characterization (substantial and challenging analytical work, also in progress). Among the current difficulties, we can list:

- Autotrophic and heterotrophic bacteria are blooming concomitantly (*Gallionellaceae* vs *Rhodocyclaceae* and unaffiliated *Betaproteobacteria* in March'12 and *Thiobacillus* vs *Desulfotomaculum* genus in May'12) and for most of the retrieved sequences, it was highly difficult to reliably infer metabolisms as those are lacking closely related described taxa with known physiologies. It is then highly difficult solely based on the available qPCR measurements to quantify in the blooms, cells that actually rely on aromatic compound degradation. In addition, metagenomic analysis clearly showed that genes related to the degradation of aromatic compounds are more abundant than genes involved in autotrophic pathways. However, it is difficult to exhaustively identify and quantify them.
- Due to the uncertainties associated with Na-flu concentration estimates for high pH and Fe-enriched groundwater, it is difficult to evaluate the quantity of Na-flu that was consumed during the survey. In addition, FT-ICR-MS results showed that the organic by-products of microbial organic matter oxidation are much more diverse than the ones deriving solely from fluorescein degradation;
- To the best of our knowledge, no rates for the abiotic PAHs' formation during basalt cooling are available in the literature. Additionally, estimating the amount of PAHs release from the rock would need to know their respective concentrations in the basalt-forming minerals and the dissolution rate of each of these minerals following CO₂ injection.

Cited references:

- Doughty, M.J. (2010) pH dependent spectral properties of sodium fluorescein ophthalmic solutions revisited. *Ophthalmic Physiol. Opt.* 30 (2), 167-174.
- Dvorski S.E.-M. et al. (2016) Geochemistry of Dissolved Organic Matter in a spatially highly resolved groundwater petroleum hydrocarbon plume cross-section. *Environ. Sci. Technol.* 50, 5536–5546.

- Einsiedl, F., Hertkorn, N., Wolf, M., Frommberger, M., Schmitt-Kopplin, P. & Koch, B.P. (2007) Rapid biotic molecular transformation of fulvic acids in a karst aquifer. *Geochim. Cosmochim. Acta* 71 (22), 5474-5482.
- Etiopie, G., Fridriksson, T., Italiano, F., Winiwarter, W. & Theloke, J. (2007) Natural emissions of methane from geothermal and volcanic sources in Europe. *J. Volcanol. Geoth. Res.* 165, 76–86.
- Geptner, A.R., Alekseeva, T.A. & Pikovskii, Yu.I., (1999a) Polycyclic aromatic hydrocarbons in fresh and hydrothermally altered volcanics of Iceland. *Dokl. Akad. Nauk*, 369 (5), 667–670.
- Geptner, A.R., Alekseeva, T.A. & Pikovskii, Yu.I. (1999b) Polycyclic aromatic hydrocarbons in volcanic rocks and hydrothermal minerals of Iceland. *Litol. Polezn. Iskop.* 6, 619–631.
- Geptner, A.R., Kristmannsdottir, H., Pikovskii, Y & Richter, B. (2005) Abiogenic hydrocarbon's emission in the modern rift zone, Iceland. *Proceedings World Geothermal Congress*, Antalya, Turkey, 24-29 April 2005.
- Gunnarsson I., Aradóttir E.S., Sigfússon B., Gunnlaugsson E. & Júlíusson B.M. (2013) Geothermal gas emission from Hellisheiði and Nesjavellir power plants, Iceland. *GRC Trans.* 37, 785-789.
- Mason, O. et al. (2010) First investigation of the microbiology of the deepest layer of ocean crust. *PLoS One* 5, e15399.
- Naim, J.O. et al. (1986) The in vitro quenching effects of iron and iodine on fluorescein fluorescence. *J. Surg. Res.* 40, 225-228.
- Rainer, U. et al. (2014) Water droplets in oil are microhabitats for microbial life; *Science* 345 (6197), 673-676.
- Scott, S., Gunnarsson, I., Arnórsson, S. & Stefánsson A. (2014) Gas chemistry, boiling and phase segregation in a geothermal system, Hellisheiði, Iceland. *Geochim. Cosmochim. Acta* 124 (1), 170-189.
- Sjöback, R., Nygren, J. & Kubista, M. (1995) Absorption and fluorescence properties of fluorescein. *Spectrochim. Acta Part A* 51, L7-L21.
- Zolotov, M.Y. & Shock, E.L. (2000) A thermodynamic assessment of the potential synthesis of condensed hydrocarbons during cooling and dilution of volcanic gases. *J. Geophys. Res.* 105 B1, 539-559.

Data availability, The MS is unacceptable until all molecular data is deposited in the appropriate publicly available data repositories.

All molecular data were deposited before initial submission in an appropriate public data repository. However the data availability statement in which all accession numbers were provided was incorrectly located in the previous version of the manuscript (after the acknowledgments, i.e. former lines 792-801). It can now be found at the appropriate location at the end of the material and methods sections (lines 675-683):

“Data availability. The authors declare that the data supporting the findings of this study are available within the article and its Supplementary Information Files. Archaea Sanger sequencing data are deposited at the GenBank of the National Centre for Biotechnology Information under accession numbers KJ867248 to KJ867371. Bacterial Sanger sequences received accession numbers from KU685482 to KU685504 and KX276716 to KX276768. Sequences from Bacteria 454-pyrosequencing and both metagenomic studies are available at the Sequence Read Archive (SRP042183, SRR3731039 and SRR3731040, respectively). Sequences reported in this paper for *cbbL*, *cbbM* genes and for genes coding for the largest subunit of multicomponent phenol hydroxylases (LmPHs) respectively received the following GenBank accession numbers: KX290744 to KX290758, KX290759 to KX290760 and KX290761 to KX290773.”

In addition, we have asked the GenBank to anticipate the release of the following references so that they are now publicly accessible:

Submission	Title	App	Status	Updated
SUB1652148	Subsurface aquifer in basaltic CCS Raw sequence reads	BioProject	BioProject: Processed PRJNA327016 : Subsurface aquifer in basaltic CCS Raw sequence reads	Sep 03
SUB1652137	Subsurface aquifer in basaltic CCS Raw sequence reads, Jun 28 '16	Sequence Read Archive (SRA)	SRA: Processed (2 objects) Download metadata file with SRA accessions Review successful SRA submissions using the SRA website	Jun 30
SUB1652152	Metagenome HN-04 March-12	BioSample	BioSample: Processed Successfully loaded SAMN05300431 : HN-04 March-12 (TaxId: 1169740)	Jun 28
SUB1652150	Metagenome HN-04 May-12	BioSample	BioSample: Processed Successfully loaded SAMN05300430 : HN-04 May-12 (TaxId: 1169740)	Jun 28
SUB515333	Subsurface aquifer in basaltic CCS Raw sequence reads	BioProject	BioProject: Processed PRJNA325213 : Subsurface aquifer in basaltic CCS Raw sequence reads	Jun 28

Specific comments.

I42, change 'could lock up' to 'have the potential to...'

This was modified accordingly (lines 49-51).

I52, presumably you mean the microbial communities living in these basalt-systems.

This sentence was rephrased accordingly (line 60).

L57, This paragraph should really start the results section, instead of being in the introduction, Fig 7 could be used here.

We do not feel comfortable with the idea of starting the result section right here because this paragraph aims at presenting, based on already published data, the framework and objectives of the study before starting the description of the results that relate to the microbiological survey of the site. Nonetheless to address Reviewer#1's concern, we moved a substantial part of this paragraph to the result section (lines

86-96) so that this section looks more like an introduction (as also requested by Reviewer#2). For former Fig. 7, as alternatively suggested below, we moved it to the Supplementary Material as new Supplementary Fig. S15. The reason lies in the fact that our study deals with groundwater samples and not with core sections, so we prefer not to use former Fig. 7 as the first figure of the paper.

I59, Why this particular combination of gases?

This combination of gases corresponds to the geothermal byproduct once the gases coming from the condensers of the power plant have been processed by the gas separation station. This is now mentioned in text (lines 70-73). Details on the separation process relying on sequential extraction are now included in the caption of Supplementary Fig. S1.

I94, Here and elsewhere, what is meant by 'fast-flowing' please provide a velocity or range of estimated velocities.

Injection well HN-02 and first monitoring well HN-04 are respectively 70 and 125 m apart at 420 and 520 m depth. At these 2 depths, feedzones displaying high density of large fractures and rubbles (and hence higher porosity compared to the massive basalt) have been identified in the aquifer and provided the fast flow pathways followed by the injected CO₂ that was detected in March 2012 in HN-04 (i.e., 60 days after the injection onset; Supplementary Figs. S2 and S15). Based on these considerations, we estimated a range of velocities that is now provided in the caption of Supplementary Fig. S2. Similarly, we assessed a range of velocities for the matrix flow which correspond to groundwater slowly circulating through the rock matrix and was later detected in HN-04 (> 200 days after the injection onset). Estimates are also included in caption of Supplementary Fig. S2 for a sake of comparison.

I142-145. As written, the meaning of this sentence is difficult to follow

This sentence was rephrased accordingly (lines 151-155).

I145, 'constantly retrieved', do you mean present in low abundance?

This was indeed the meaning of the statement. The sentence was rephrased to better reflect it (lines 158-160): "Betaproteobacteria, already present in low abundance prior to injection, became dominant after, accounting for 87.5-87.8% and 92% of the 16S-rRNA gene sequences obtained by pyrosequencing and metagenomic analysis, respectively (Figs. 1f, 2b, 3, S10)."

I148, more resistant? Or was a subset selected for growth by the changed conditions?

We agree that it is more appropriate to talk about a subset that was selected for growth by the changed physical and chemical conditions. The text was changed accordingly (lines 160-162): "Decrease in pH and rise in metal concentrations (e.g. Zn concentrations increased by a factor of 520; Table S3) fostered proliferation of this population as supported by statistical analysis (Fig. 5; Table S5)."

I162, Here and elsewhere, according to Linnean principles, only genus and species are italicized.

This was changed throughout the manuscript and in Supplementary Material. Figs. 1 to 3 and Fig. 7 were also modified accordingly.

Paragraph starting with I153. Please see the paper of Barco et al, AEM 2015 regarding the cyc2 gene, which is present in Sideroxydans and for which there is evidence of it being involved in Fe-oxidation. The authors need to specifically determine if this cyc2 homolog from Sideroxydans is present in their dataset, also see the recent paper by Beller et al ISME J 2016, where evidence for Gallionellales was found in an aquifer, and a cyc2 homolog was abundantly expressed.

We thank Reviewer#1 for her/his suggestion. As both Pfam motifs and COG databases are devoid of any relevant searchable entry, determination of *cyc2* gene homologs differential expression was performed by direct sequence similarity search of 2 *cyc2* homologs namely AKN78226.1 (*Cyc2PV-1* from *Mariprofundus ferrooxydans*) and ADE10507 (*Slit_0265* from *Sideroxydans lithotrophicus*), against the appropriate full reads datasets.

Table below shows hits matching both sequences in the metagenomic data obtained on groundwater samples collected in March and May 2012. The high number of hits for *Slit-0265* unambiguously confirmed the overrepresentation of iron oxidizers in March 2012 in the aquifer, in agreement with the bloom of species affiliated to *S. lithotrophicus*. We amended the text with these results (lines 186-192) and both papers mentioned by Reviewer#1 were added to the reference list. These data are also shown in the new Supplementary Table S8.

	hits in March 2012	hits in May 2012	Normalized number of sequences for March 2012	Normalized number of sequences for May 2012
ADE10507:Slit_0265	2495	129	0.5684	0.0319
AKN78226.1:Cyc2PV-1	341	85	0.0777	0.0210

I 170, acidothiobacilli is an anachronistic term, better to say strains related to *Acidithiobacillus*.

This was modified accordingly (lines 181-182).

I174, It's not clear what the point is here, the conversion of DIC to biomass must be very small relative to the amount of CO₂ that is being introduced into the system. A better accounting or approximation of how much biomass may have actually resulted from this process would be useful.

*As previously discussed for heterotrophic metabolisms that rely on aromatic compound degradation, it is similarly highly difficult to assess how much biomass may have resulted from the autotrophic conversion of DIC. The first difficulty hampering the estimates relies on the fact that autotrophic and heterotrophic bacteria are blooming concomitantly and that for most of the retrieved sequences, it is difficult, if not impossible, to reliably infer metabolisms as those are lacking closely related described taxa with known physiologies. In the present case, qPCR measurements can accordingly only provide an estimate of the global bacterial bloom but not of the respective metabolisms. In addition, some of the injected CO₂ could have been used not only by autotrophs but also by heterotrophs. First, as observed in a wide variety of habitats, from oil reservoirs and coal deposits to contaminated groundwater and deep sediments, CO₂ can be used as electron acceptor to sustain intermediate steps in the microbial transformation of hydrocarbons (Jiménez et al., 2016). Second, representatives of the *Desulfotomaculum* genus, as the species blooming in May 2012, assimilate for biomass production, Acetyl-CoA through reductive carboxylation. They hence consume environmental DIC and high ratios of CO₂ was shown to be fixed by these heterotrophs through this carboxylation reaction (Winderl et al., 2010; Taubert et al., 2012). A strain of *Desulfotomaculum* was isolated from the *Hellisheidi* groundwater sampled during the present survey. Its genome was sequenced and is currently examined to further document its metabolic capabilities.*

*We rephrased the sentence to more clearly express our aims (i.e. not quantifying but pointing to enhanced autotrophic pathways in the aquifer for March 2012 through the presence of 16S rRNA gene sequences affiliated to autotrophic species (i.e. *Gallionellaceae* members affiliated to *Sideroxydans lithotrophicus* species) combined with the concomitant selective amplification of *cbbM* genes in addition*

to *cbbL* genes): “The bloom of Gallionellaceae members is indicative of an increased potential for autotrophic C-fixation and thereby CO₂ conversion into biomass in the aquifer in March’12.” – (lines 196-197).

Cited references:

Jiménez, N., Richnow, H.H., Vogt, C., Treude, T. & Krüger, M. (2016) Methanogenic hydrocarbon degradation: evidence from field and laboratory studies. *J. Mol. Microbiol. Biotechnol.* 26 (1-3), 227-42.

Taubert, M. et al. (2012) Protein-SIP enables time-resolved analysis of the carbon flux in a sulfate-reducing, benzene-degrading microbial consortium. *ISME J.* 6, 2291-2301.

Winderl C., Penning H., von Netzer F., Meckenstock R.U. & Lueders T. (2010) DNA-SIP identifies sulfate-reducing Clostridia as important toluene degraders in tar-oil-contaminated aquifer sediment. *ISME J.* 4, 1314-1325.

I221-226. The writing is very awkward here, so it's difficult to follow the point they are making.

This sentence was rephrased (lines 245-251).

I230-231, This is really a discussion point.

To address Reviewer#1's concern, this point was moved to the discussion (lines 324-326).

I231, conversely, poor word choice, either, At the same time, or Concurrently, would be better.

All the paragraph was rewritten.

I238-240, It's not clear what is meant here, T.d and S.l share little overall gene homology, so either their analysis is incorrect, or they are discussing a specific gene or genes where these organisms may show more homology.

We agree that the two species share little overall gene homology and the statement only concerned a specific gene cluster (i.e. the soxXYAB gene cluster in Sideroxydans ES-1 that shares significant homology and gene order with the soxXYAB gene in Thiobacillus denitrificans). This is now more clearly indicated (lines 264-267).

pg13, I'm missing the importance of Fig 5 here that shows most genes that except for acetyl Co-A genes, are expressed in the same range or more highly expressed in March compared to May, suggesting the system was already primed for the biomass bloom that they saw in May. This does not seem to be consistent with their explanation of a 'bloom' in May, especially of aromatic degradation.

We agree that the major differences between the key biomarkers retrieved in HN-04 in May and March 2012 are only threefold: (i) an increase in markers of the oxidative acetyl co-A pathway and of the yedY encoding genes, (ii) a greater potential for the use of S-compounds (with increased aprA, aprB, dsrAB markers mediating dissimilatory sulfate reduction) and (iii) a higher signal intensity of the nirK markers indicative of assimilatory N reduction. With the exception of the yedY marker that can be widely distributed among various phyla, all these markers may all be related to the bloom of Firmicutes. First, the observed enhancement of N assimilation could be due to the increase of microbial biomass largely composed of Firmicutes in May 2012 (Hazen et al., 2010). Second, the anaerobic degradation of aromatic compounds that can be coupled among the Firmicutes to the reduction of S-bearing compounds, ultimately yields acetyl-CoA that in turn can be oxidized by means of the bifunctional CODH/acetyl-CoA synthase to produce ATP. This latter was shown to be used to initiate ring cleavage associated with the ATP-dependent degradation of aromatic compounds (Bozinovski et al., 2014).

The lack of contrasted distributions for the carbon-related markers between March and May 2012, as observed in former Fig. 5 (now Fig. 6), likely result from the fact that for both seasons, autotrophs and aromatic hydrocarbon degraders are blooming concomitantly (Gallionellaceae vs Rhodocyclaceae and

unaffiliated Betaproteobacteria in March 2012 and Thiobacillus sp. vs Desulfotomaculum sp. in May 2012). However, the distribution of the carbon-related markers is in agreement with the gene sequences coding for the largest subunit of multicomponent phenol hydroxylases involved in phenolic compound degradation that were successfully amplified only in March and May 2012. The potentiality of the Hellisheidi ecosystem to degrade PAHs as observed in March and May 2012 strongly suggests that the community is used to PAHs as a carbon source. If we consider, as proposed above, that aromatic compound degradation could correspond to a characteristic feature of microbial communities inhabiting mafic environments, the system should indeed be already primed to react to an increase of PAHs, as likely provoked here by the dissolution of the host basalt. As now shown by Mantel tests shown in new Supplementary Table S5, the way the community was differentially shaped between March and May 2012 may be more likely linked to the different chemical conditions that prevailed in the aquifer at both sampling periods (i.e. pH, DIC, DOC, metal concentrations along with O₂ fugacity). pH and Fe, Mg, Ca temporal evolutions that are now reported in Supplementary Fig. S2 show that, compared to March 2012, the aquifer present in May 2012 geochemical parameters that are closer to the pre-injection ones. This may have allowed the aquifer community to flourish again under less acidic conditions.

The discussion was significantly rewritten to make this point clearer.

Cited references:

- Bozinovski, D. et al. (2014) Metaproteogenomic analysis of a sulfate-reducing enrichment culture reveals genomic organization of key enzymes in the m-xylene degradation pathway and metabolic activity of proteobacteria. *Syst. Appl. Microbiol.* 37, 488-501.
- Hazen et al. (2010) Deep-sea oil plume enriches indigenous oil-degrading bacteria. *Science* 330 (6001), 204-208.

I253-256 Protein motif is incorrect, this work is based on metagenomics not proteomics.

As we are dealing here with metagenomic sequences, sequence motifs such as Pfam motifs (which are short statistical sequence motifs, as opposed to sequence similarity and contrary to structural motifs) are quite appropriate to describe our data-mining approach. We replaced “protein motifs” with “protein sequence motifs” throughout the text to comply with the Reviewer#1’s request.

I256, what is an aromatic cycle?

We changed “cycle” into “ring”, of more common use (line 283).

I262-263, More unspecific text, what's meant by 'some of the importantly'

We now provide percentages (lines 286-291): “At the same time, pyrosequencing and metagenomic analysis showed that Rhodocyclaceae were still dominant (respectively representing 12.2-12.9% of the pyrosequences and 27% of the 16S-rRNA gene sequences in the metagenomic data) and that Ignavibacteriales, Sphingomonadales and OD1 Parcubacteria, originally inhabiting the aquifer, were retrieved again (representing respectively 11.0-12.4%, 2.1-3.7% and 5.3-5.4% of the pyrosequences and 6%, 7% and 1% of the 16S-rRNA gene sequences in the metagenomic data).”

I264-265, It's not clear what is meant by 'close arrival of the fast-flowing fraction...'

We rephrased the sentence (lines 292-294): “In July’12, temporal evolutions of non-reactive SF₅CF₃ tracer, DIC and H₂S concentrations indicated that the fast-flowing fraction of the geothermal gas mixture had reached HN-04 (Fig. S2a,e; Table S1).”. In order to better illustrate the timing of the arrival of the geothermal mixture at the level of monitoring well HN-04, we also added in Supplementary Fig. S2, the temporal evolution of H₂S concentrations in HN-04 groundwater.

I288, resistance would be a difficult thing to prove in this case, better to say adapted or more adaptable

All the section was rewritten. and we now avoid the use of the word “resistant” throughout the text.

1st paragraph of discussion. It’s curious they begin the discussion here. This argument for micro-nutrient availability only works if the system is limited for micro-nutrients rather than electron donor or acceptor. In general rock-hosted systems are not thought to be limited for mineral-based micronutrients, except perhaps N; they provide no references to support their case. Because they do not provide any information on actual concentrations of either electron donors or acceptors in the system the argument that this is a micro-nutrient limited system is very weak and should be re-considered.

Because the central question of the discussion is the source of carbon that sustained the microbial blooms (now more thoroughly addressed), we chose to start the discussion with the potential role of metals (as both nutrients or energy sources in the case of Fe) which were released from the basalt following the acidification by the CO₂-charged groundwater (Supplementary Table S3 and new Fig. S2). While not claiming for oligotrophy in the aquifer, we found striking the correlation between the series of metals whose concentrations were enhanced following basalt dissolution and the metallic cofactors involved in the key markers highlighted by metagenomic analysis at the same time periods (especially the ones that relate to PAH degradation). Before the CO₂ injections and compared to May 2012, all these metal concentrations were significantly lower in groundwater (Supplementary Table S2) as was the total biomass (Fig. 1e). In the particular case of iron, we observed an increase of more than 3 orders of magnitude (from few ppb to more than 1100 ppb, Supplementary Table S3 and Fig. S2d) pointing to a potential role of the metal released from the basalt in the stimulation of iron-oxidizers belonging to the Gallionellaceae family. Our aim was accordingly to highlight the link between host rock dissolution and enhanced metabolisms. We now provide the temporal evolutions of Fe, mainly in the form of Fe²⁺, Mg, Ca along with total S, SO₄²⁻ and H₂S to better support the discussion. We also removed in the introduction the misleading statement on oligotrophy in order to avoid suggesting such a context (former line 51).

I315 - 317. Should say something about the fluorescein concentrations and dilution, is it reasonable to assume there was enough organic carbon in the fluorescein to support significant growth?

In addition to absolute concentrations now provided lines 115-117, 237 and 297, we added to the main text (lines 115-117) the range of increase in Na-fluorescein concentrations (i.e. x20). Full temporal evolution can also be found in Supplementary Fig. S2c.

Na-flu concentrations were likely not sufficient to support significant growth. In addition, the degradation of this compound would not generate by itself the organic molecular diversity observed by FT-ICR-MS (Supplementary Fig. S4). As discussed above, the various sources of aromatic compounds that could have sustained the microbial blooms are now more thoroughly discussed (lines 362-404).

I331-333, clogging discussion, it seems another likely clogging culprit was the bloom of Sideroxydans, FeOB are common problems in water aquifers due to clogging.

We thank Reviewer#1 for her/his suggestion. We amended the text and the reference list to mention this point (lines 362-362).

I384-386, meaning of sentence is unclear as written.

The sentence was rephrased accordingly (line 451-454).

I390, please provide depth values.

Depth values for the pumps and the casing in HN-01 and HN-04 are now provided (lines 456-460).

I506-507, why not just briefly outline the quality control process rather than saying it was 'rigorous'?

The quality control process was detailed after this statement (former lines 507-511): “Sample barcode and primer sequences were trimmed from the proximal end of the sequences without any truncation of the sequences on the 3’ end, as recommended for proper use of Denoiser⁶³. Reads shorter than 300 nucleotides or containing one or more ambiguities were removed. This process was done twice in order to apply Denoiser on both the forward and the reverse set of sequences”.

We changed the sentence introducing the quality control process with: “Reads were quality filtered by removing reads with unresolved bases and/or anomalous read length” (lines 577-578).

I590. Data availability, The MS is unacceptable until all molecular data is deposited in the appropriate publically available data repositories.

Please see our previous answer above.

Fig 6 legend and elsewhere, it's unclear what is meant by inhibition, since they did not track individual populations how is it know whether this is inhibition, or simply outgrowth of specific populations.

We agree that the word “inhibition” was not appropriate. The sentence has been rephrased accordingly (line 1009 to 1012): “In March 2012 (Mar’12), groundwater acidification induced by arrival of the fast-flowing fraction of the pure-CO₂ injectant severely reduced bacterial richness, provoked the dissolution of the host-basalt and the release of mineral-forming cations and likely polyaromatic hydrocarbons (PAHs).”.

No other similar occurrences were found.

Figure 7, It seems like this figure should be figure 1, since it sets the physical setting. However, it might best as supplemental. I cannot make out what meant in 7e and the explanation does not help. The Fe-oxide slimes are interesting, FeOB are known for producing unique morphological structures, they should state whether or not any of these were observed by microscopy.

Because our study dealt with groundwater samples and not core sections, we prefer not to use former Fig. 7 as the first figure of the paper. As alternatively suggested, we moved Fig. 7 to the supplementary material (this latter figure is now numbered Supplementary Fig. S15). To clarify former panel 7e whose aim was to show that vesicles were sometimes unusually large as the one now shown by an arrow in the figure, we rephrased the caption: “Vesicles were sometimes unusually large as the one shown by a white arrow on the core section retrieved at 479.09 m depth (e).”

The observations by optical or scanning electron microscopy of the Fe-oxides slimes covering the core fracture surface or filling the vesicles did not revealed distinctive morphologies such as extracellular twisted ribbon-like stalks that could have been attributed to Fe-oxidizing bacteria. Instead of twisted iron stalks, we found interspersed and layered iron oxides and clays, both intimately mixed. Those can nonetheless also constitute byproducts of FeoB metabolisms because several FeOB including Sideroxydans sp., and Gallionella capsiferiformans are known to produce amorphous iron oxides (Emerson and Moyer, 1997, 2002; Weiss et al., 2007; Henri et al., 2016). As a first clue of the presence of iron-oxidizing bacteria, we found cells entombed in iron-oxides and clays by Transmission Electron Microscopy carried out on ultrathin sections milled by Focused Ion Beam but the characterization at the micro- and nano-scale of these assemblages is still in progress.

Cited references:

Emerson D., Moyer C. (1997). Isolation and characterization of novel iron-oxidizing bacteria that grow at circumneutral pH. *Appl Environ Microbiol* 63, 4784–4792.

Emerson D., Moyer C. L. (2002). Neutrophilic Fe-oxidizing bacteria are abundant at the Loihi Seamount hydrothermal vents and play a major role in Fe oxide deposition. *Appl Environ Microbiol* 68, 3085–3093.

Henri P. A., Rommevaux-Jestin C., Lesongeur F., Mumford A., Emerson D., Godfroy A. and Ménez B. (2016) Structural iron (II) of basaltic glass as an energy source for Zetaproteobacteria in an abyssal plain environment, off the Mid Atlantic Ridge. *Front Microbiol* 6, 1518. doi: 10.3389/fmicb.2015.01518

Weiss J. V., Rentz J. A., Plaia T., Neubauer S. C., Merrill-Floyd M., Lilbur T. (2007). Characterization of neutrophilic Fe (II)-oxidizing bacteria isolated from the rhizosphere of wetland plants and description of *Ferritrophicum radicola* gen. nov. sp. nov., and *Sideroxydans paludicola* sp. nov. *Geomicrobiol. J.* 24, 559–570.

Reviewer #2 (Remarks to the Author):

The language is generally good but there are a few stylist points that could be improved. I have noted some of these in the minor comments but the manuscript would benefit from proofreading.

We thank Reviewer#2 for his/her help in improving the manuscript. The paper was proofread by a native English colleague in order to improve poor stylistic points. We now hope that it fulfills all language criteria.

I realize that much of the information is published, but the text from line 57 to 81 feels like methods. Could this be edited to be more of an introduction?

This section was rewritten to address Reviewer#2's concern. Some of the information was moved to caption of Supplementary Fig. S1 or to the Result section (lines 86-96).

In general, the manuscript would be improved throughout by replication of data and addition of statistics and significant correlations between the tested parameters. The first example is the need to add standard deviations data from Table S2 to Fig. 1 and the text in lines 96-100 and then state where the differences were significant. In the same manner, please present statistics for ion release (line 102) and statistically significant differences presented with number of replicates, SDs or confidence limits. The 16S rRNA gene community profile would also benefit from replication and statistics. This would show if the differences between HN-01 and HN-04 were significant.

In order to assess if the differences between the 2 monitored wells are significant, we acknowledge that statistical criteria are critical.

- *Although it was previously indicated in Fig. 1 and associated caption that numbers of OTU and Shannon index were both obtained in duplicates from 454-pyrosequencing of 16S-rRNA amplicons, we now present replicated data for the 16S rRNA gene community profile (new Supplementary Fig. S5). It shows that the taxonomy at the species level is similar and that the relative proportions of the obtained taxa are in good agreement for the two duplicates for both HN-01 and HN-04 wells. Relative proportions of OTU now include the values of both replicates when cited in the text;*
- *We also provide statistics for the 16S rRNA gene community profile including Mantel tests (Diniz-Filho et al., 2013) and canonical correlations based on variance analysis (new Fig. 5 and new Supplementary Table S5). They show in particular for March 2012, a strong correlation between iron concentrations in groundwater and the OTU blooming at that sampling time;*
- *Standard deviations were already provided in Tables S2 and Table S3. For Fig. 1 the errors for pH and conductivity measurements as formerly indicated in the material and methods (section "Microbiological sampling procedure") were smaller than symbol size. This is now clearly indicated in caption of Fig. 1. For the measurements that were not carried out in the framework of the present study, we now provide uncertainties on measurements when available in the literature (e.g. uncertainties on calculated DIC measurements were estimated to be $\pm 5\%$ by Matter et al., 2016);*

- All qPCR measurements were done in duplicates as indicated in the Material and methods section;
- For statistical analysis shown in Fig. 4, numerical uncertainties based on 1,000 replicates were already shown and explained in associated caption;
- We amended Fig. S2 with temporal evolution of pH, dissolved organic carbon, Fe, mainly in the form of Fe²⁺, Mg, Ca along with total S, SO₄²⁻ and H₂S concentrations that were recently published in Snæbjörnsdóttir et al. (2017). They help in assessing how significant the chemical changes were. Uncertainties on most of these measurements can be found in the cited reference.

Cited references:

Diniz-Filho, J.A., Soares, T.N., Lima, J.S., Dobrovolski, R., Landeiro, V.L., de Campos Telles, M.P., Rangel, T.F., Bini, L.M., 2013. Mantel test in population genetics. *Genet. Mol. Biol.* 36, 475e485.

Matter, J.M. et al. Rapid carbon mineralization for permanent disposal of anthropogenic carbon dioxide emissions. *Science* 352, 1312-1314 (2016).

Snæbjörnsdóttir, S. Ó. et al. CarbFix: The chemistry and saturation states of subsurface fluids during their situ mineralisation of CO₂ and H₂S at the CarbFix site in SW-Iceland. *Int. J. Greenh. Gas Control* 58, 87-102 (2017).

Why were genes involved in the CBB cycle targeted for carbon dioxide fixation (lines 135)? This is the most “energy expensive” method for carbon fixation and genes for other carbon fixation processes should also be investigated. In fact, the authors later discuss the reductive carboxylate cycle and this should also be included.

We tried to selectively amplify, in addition to the cbbL and cbbM genes respectively encoding form I and II RuBisCO, the other genes involved in autotrophic C-fixation. Those included:

- *accA (with the primers Crena_529F / Crena_981R), encoding the acetyl-CoA carboxylase alpha subunit involved in the 3-hydroxypropionate/4-hydroxybutyrate (HP/HB or 3HP/4HB) cycle;*
- *acIB (with the primers 892F / 1204R), encoding Beta ATP citrate lyase involved in the reductive Tricarboxylic Acid cycle;*
- *although the pcs gene encoding the propionyl-CoA synthase (pcs) was not amplified, we tested mcrA (with the primers MLF / MLR) as part of the genes involved in the 3-Hydroxypropionate (3HP) bicycle*
- *acs (with the primers ACS-f / ACS_r) encoding acetyl-CoA synthase for the reductive acetyl CoA pathway*

All the PCR experiments targeting these latter genes were unsuccessful. The unsuccessful PCR amplifications are in agreement with the metagenomic data that did not highlight any other CO₂ fixation pathway in addition to the CBB cycle (as now more clearly indicated lines 207, 256-257). Indeed, we only detected genes coding for the phosphoribulokinase (prkB) and subunits of ribulose-1,5-bisphosphate carboxylase (rbcL and rbcS, respectively). Note that, as indicated in Fig. 6, Supplementary Table S7 and in text (lines 281-283), the key markers retrieved by metagenomic analysis point to the oxidative Acetyl-CoA pathway instead of the reductive one.

Although representing the most “energy expensive” method, it has been shown that both RubisCO forms I and II allowed efficient CO₂ fixation in environments with strong fluctuations of O₂ and CO₂ concentrations (Herrmann et al., 2015), as it was the case in the Hellisheidi aquifer, especially during the CO₂ injections. Most of the cbbL and cbbM phylotypes detected in Herrmann et al. (2015) were closely related to Sideroxydans sp. . cbb is known to be used by Sideroxydans and Thiobacillus species for carbon fixation (Emerson et al., 2013). Notably, Sideroxydans sp. possesses the gene to encode form II RuBisCO which was solely detected during the bloom of those iron-oxidizers (i.e. in March 2012; Supplementary Fig. S8).

Cited references:

Berg, I.A. et al. (2010) Autotrophic carbon fixation in archaea. *Nat. Rev. Microbiol.* 8, 447-460.

Emerson, D. et al. (2013) Comparative genomics of freshwater Fe-oxidizing bacteria: implications for physiology,

ecology, and systematics. *Front. Microbiol.* 4, 254.

Herrmann, M. et al. (2015) Large fractions of CO₂-fixing microorganisms in pristine limestone aquifers appear to be involved in the oxidation of reduced sulfur and nitrogen compounds. *Appl. Environ. Microbiol.* 81, 2384–2394.

Has HPIP been detected in other iron oxidizing populations than the acidithiobacilli (line 169)? Otherwise this would be a surprising finding as species in this genus are obligate acidophiles.

According to Bird et al. (2011) and Bonnefoy and Holmes (2012), although data are still scarce, 2 operons for iron oxidation have been described for neutrophilic species. One of them, the pioABC operon in Rhodospseudomonas palustris TIE-1 encodes a decahem periplasmic cytochrome c, an outer membrane protein and a high HiPIP protein (Jiao and Newman, 2007) and has been suggested to be involved in iron oxidation. Nonetheless, the other protein-sequence motif which was also abundantly retrieved in March 2012 related to PhnJ, an ABC transporter for phosphonate uptake. PhnJ is not specific to Gallionellaceae and has been found only in acidophilic Sideroxydans strains (Mühling et al., 2016). Canonical correlations also suggested the community present in March 2012 in the aquifer may have a higher tolerance to lower pH (Fig. 5). This is now mentioned lines 184-186.

Cited references:

Bonnefoy V., Holmes D. S. (2012) Genomic insights into microbial iron oxidation and iron uptake strategies in extremely acidic environments. *Environ Microbiol* 14 (7), 1597-611.

Bird L. J., Bonnefoy V., Newman D. K. (2011) Bioenergetic challenges of microbial iron metabolisms. *Trends Microbiol* 19 (7), 330-40.

Jiao Y., Newman D. K. (2007) The *pio* operon is essential for phototrophic Fe(II) oxidation in *Rhodospseudomonas palustris* TIE-1. *J Bacteriol* 189, 1765–1773.

Mühling M., Poehlein A., Stuhr A., Voitel M., Daniel R., Schlömann M. (2016) Reconstruction of the metabolic potential of acidophilic *Sideroxydans* strains from the metagenome of an microaerophilic enrichment culture of acidophilic iron-oxidizing bacteria from a pilot plant for the treatment of acid mine drainage reveals metabolic versatility and adaptation to life at low pH. *Front. Microbiol.* 7, 2082.

Could the source of aromatic carbons be from e.g. humic acids from the photosynthetically fed surface, from the Na-fluorescein (line 198), or drill fluids? Is it known how long the water in the present system has been partitioned from the surface and which if any of these possibilities is the most likely?

We thank Reviewer#2 for her/his suggestion. We fully agree that the source of the organic matter sustaining the presence of genes involved in the degradation of aromatic compounds is a critical point, as also raised by Reviewer#1. We drafted a discussion on its likely origins in the former version of the manuscript while acknowledging however that these sources were not well constrained and difficult to identify (former lines 313 to 323). We now elaborated on this point (from lines 362 to 404) by further discussing the FT-ICR-MS data shown in Supplementary Figs. S3 and S4 and by introducing insights from literature.

*As discussed in the previous version of the manuscript, the first evident and likely source is indeed the **Na-fluorescein** (Na-flu), a biodegradable heterocyclic compound. To better compare its increase in HN-04 groundwater with the amplitude of the bacterial bloom, Na-flu concentrations are now more clearly provided (lines 115-117, 237, 297) as a complement to its temporal evolution shown in Supplementary Fig. S2. Note nonetheless that those could have been underestimated as Na-flu absorbance measurements could have been hampered by groundwater pH values and elevated Fe concentrations, especially in March 2012 where iron showed its highest concentrations (Naim et al., 1986; Sjöback et al., 1995; Doughty, 2010). Anyhow, while bacteria may have benefited from this source of carbon, the x20 increase in Na-flu concentrations may difficultly explain by itself the x500 increase in biomass observed in May 2012. In addition, the degradation of Na-Fluorescein would not generate the organic molecular diversity established by FT-ICR-MS and shown in Supplementary Fig. S4. Consequently we may postulate*

that additional sources of organic carbon would have been available in the aquifer during the monitored period.

Alternatively, as also suggested by Reviewer#1, we then considered the following sources as potential candidates. Among them are, as suggested by Reviewer#2:

- **humic acids from the photosynthetically fed surface:** the van Krevelen distribution of the organic matter found all along the survey (Supplementary Fig. S4) shows rather aliphatic-type structures not typical of groundwater organic matter/humic acids that are more aromatic and highly oxidized (Einsiedl et al., 2007). Thus, this limits the hypothesis of a photochemically fed surface origin for the organic compounds. Supplementary Fig. S4 was modified accordingly to display the distribution expected for dissolved organic carbon derived from the surface biomass and now clearly shows the discrepancy;
- **drilling fluids:** injection and monitoring wells were drilled using groundwater from the aquifer as drilling fluids, thus implying that no organic muds were added to the system. If a contribution is expected from drilling fluids, it then corresponds to endogenic organic compounds coming from the deep basaltic aquifer itself (please see below);

From that perspective, based on additional clues from the FT-ICR-MS data and insights from literature, we now consider that **endogenic organic compounds coming from the rock itself** can constitute a valuable source of carbon that could have been released during basalt dissolution. This is now developed in the manuscript (from lines 362 to 404). We also modified caption of Supplementary Fig. S4 in order to better discuss the likely sources of organic compounds.

Cited references:

- Doughty, M.J. (2010) pH dependent spectral properties of sodium fluorescein ophthalmic solutions revisited. *Ophthalmic Physiol. Opt.* 30 (2), 167-174.
- Einsiedl, F., Hertkorn, N., Wolf, M., Frommberger, M., Schmitt-Kopplin, P. & Koch, B.P. (2007) Rapid biotic molecular transformation of fulvic acids in a karst aquifer. *Geochim. Cosmochim. Acta* 71 (22), 5474-5482.
- Naim, J.O. et al. (1986) The in vitro quenching effects of iron and iodine on fluorescein fluorescence. *J. Surg. Res.* 40, 225-228.
- Sjöback, R., Nygren, J. & Kubista, M. (1995) Absorption and fluorescence properties of fluorescein. *Spectrochim. Acta Part A* 51, L7-L21.

Why were the metagenomes not binned to “metagenome assembled genomes”? This would have allowed metabolic functions suggested by the identified genes to be ascribed to phylogenetic taxa. It would also alleviate the point below.

Metagenomes binning is slightly off-topic in the context of this work (but further work is in preparation). This work focuses primarily on comparing metabolic output of the existing microbiome, not on the compositional evolution thereof. Moreover, we believe that the phylogenetic diversity of the studied biotope is sufficiently high to deserve a specific publication

In general, please be careful in assigning metabolic functions to populations based on 16S rRNA gene alignments. There are many cases where high 16S gene similarities do not correspond to similar functions. The results and discussion should be edited to reflect this limitation.

We acknowledge that the assignment of metabolic functions only based on the 16S-rRNA gene alignment can largely be prone to misinterpretations. That is why we very carefully limit our assumptions on the functioning of the ecosystems in terms of electron donors and acceptors, with the exception of the members of the Gallionellaceae family for which all cultivated members characterized so far are chemolithoautotrophic microaerophilic iron-oxidizing bacteria. We took care to this point in editing the result and discussion sections.

Lines 393-400: What was the volume of water needed to be pumped to ensure that the stagnant well water was removed from HN-04 and HN-01? Based on these numbers, how many volumes were removed by pumping 8.6 E 4 litres and 2.5 E 5 litres from HN-04 and HN-01? The actual amount pumped is not the critical point; it is that the pumped volume should be at least three times the stagnant well water volume.

Typically in the wells drilled at Carbfix1 site, the volume of the anchor casing and of the inner pipe were <1.9 m³ and <1.2 m³, respectively (i.e. 3.1 m³ of stagnant water). Three times the stagnant well water volume corresponds to less than 10 m³. The actual pumped volume before sampling was well above this critical point (respectively 8x and 25x the stagnant water volume for HN-04 and HN-01) thus ensuring that the samples represented fresh groundwater rather than stagnant well water. We completed accordingly the "Microbiological sampling procedure" paragraph in the Material and Methods section.

Lines 401-403: I realize that the samples were taken several years ago, but recent studies have shown ultrasmall cells in oligotrophic groundwaters that will (at least partially) pass through a 0.22 µm filter (see: Luef et al (2015) Nat Commun 6:doi: 10.1038/ncomms7372). It would be interesting to test for small cells and this could be commented upon.

Based on the recent work of Luef et al. (2015), ultrasmall cells could have been indeed relevant to characterize. Unfortunately, groundwater were filtered directly on the field by fixing sterile 0.22 µm Sterivex™-GP filter units directly on the well head and only the Sterivex were kept at -20°C until DNA extraction.

Lines 420-422: Did the authors test for amplification with archaeal 16S primers? If not, this should be noted and the text altered to reflect the qPCR abundance estimates reflected the total bacterial and not prokaryotic abundance.

We indeed ran qPCR analysis with archaeal 16S primers. However, due to their low abundance in March and May 2012, as highlighted by metagenomic analysis (Supplementary Fig. S10), those data were not shown in the previous version of the manuscript. They are now included as Supplementary Fig. S14. We also completed the Material and Methods Section and Table S9 along with the Supplementary reference list with information on qPCR experiments performed on archaea. We agree that the abundance shown in Fig. 1 only reflects the total bacteria and not the total prokaryotes as previously indicated in the associated caption.

Line 517: Why were "sequences non-affiliated to a phylum" removed? Would this not have removed any novel candidate phyla identified in the samples?

We mistakenly wrote phylum and apologize for this inconsistency. It was the sequences that were not affiliated to an OTU (i.e. the singletons) that were removed. As it can be seen in Figs. 2 and 3 where they are named "unknown Bacteria" or "unknown sequences", sequences non-affiliated to a phylum are still present in the dataset and actually represent a large portion of the retrieved sequences. We fixed this in text lines 588-589.

Minor comments

Line 27: I suggest: "In this study, we have carried out the first..."

This was modified accordingly (lines 32-33).

Line 28: I am not sure what a "high" response is. Please give a quantitative measurement?

We rephrased the sentence to describe changes more accurately (lines 33-37): "[...] we show that deep

ecosystems respond quickly to field operations associated with CO₂ injections. Acidic CO₂-charged groundwater results in a marked decrease (by ~2.5-4) in microbial richness despite observable blooms of lithoautotrophic iron-oxidizing betaproteobacteria and degraders of aromatic compounds [...].

Line 31: "... resulted in a marked decrease in microbial richness."

Please see above.

Line 37: This is a long sentence with many clauses. Please split into two to simplify the language.

The sentence and the paragraph were modified to address both Reviewers' concerns (lines 42-51). We hope that these changes make the reading more linear.

Line 37: "...would reconcile the use..."

This was modified accordingly (line 43).

Line 40: Once again, the sentence has many clauses and could be simplified. I will not comment on this again but please read through the manuscript and edit further instances.

The sentence was simplified here and elsewhere when sentences had too many clauses.

Line 55: Replace "retroactively" with "subsequently"?

Line 75: "...study focuses on..."

Line 76: "... which a few percent..."

Changes were made accordingly (lines 62, 91 and 92).

I will not make language style suggestions after this point, but feel the text could be proofread.

We thank Reviewer#2 for her/his help in improving the readability of the manuscript. The paper was proofread by a native English colleague. We now hope that it fulfill all language criteria.

Line 130: Was the pH drop significant? Please support the text with data.

The pH dropped by ~ 0.6 pH unit. This is now clearly indicated in the main text (line 138).

Line 143: Please avoid adjectives like "drastic". Instead, please provide numbers and statistics to support your statements.

We removed "drastic" and recalled the amplitude of pH drop and conductivity increase. For the microbial community, changes are then thoroughly described in the paragraph which follows this statement

Line 148: I would not say that Betaproteobacteria "appeared more resistant to the pH decrease and the rise in metal concentrations" but rather these conditions allowed these populations to proliferate.

We agree and modified the sentence accordingly (lines 160-162): "Decrease in pH and rise in metal concentrations (e.g. Zn concentrations increased by a factor of 520; Table S3) fostered proliferation of this population as supported by statistical analysis (Fig. 5; Table S5)."

Line 165: "cytochrome"

Line 167: The c in "c-type" should be in italics.

Both typos are now fixed (line 176 and 179)

Line 224: I would suggest the increase in cell numbers was "likely" due to the carbon dioxide injection.

“Likely” was added to the sentence (line 250).

Line 273: The data suggest that the conditions did not favor autotrophic species that fixed carbon via the CBB cycle. Other carbon fixation systems should also be investigated.

Before the CO₂ injections, no or very few potentially-autotrophic strains were detected. In addition, the cbbL and cbbM genes respectively encoding form I and II RuBisCO involved in the CBB cycle were only amplified from the groundwater collected in February, March and May 2012, i.e. while the injected gas was travelling in the aquifer. Consequently, autotrophic species that use the CBB cycle were only fostered within the plume of CO₂.

Although the CBB cycle was the only pathway of CO₂ fixation that was detected in the metagenomic data, we tried to selectively amplify, in addition to the cbbL and cbbM genes, other genes involved in autotrophic C-fixation. All were however unsuccessful and included:

- *accA (with the primers Crena_529F / Crena_981R), encoding the acetyl-CoA carboxylase alpha subunit involved in the 3-hydroxypropionate/4-hydroxybutyrate (HP/HB or 3HP/4HB) cycle;*
- *aclB (with the primers 892F / 1204R), encoding Beta ATP citrate lyase involved in the reductive Tricarboxylic Acid cycle;*
- *although the pcs gene encoding the propionyl-CoA synthase (pcs) was not amplified, we tested mcrA (with the primers MLF / MLR) as part of the genes involved in the 3-Hydroxypropionate (3HP) bicycle*
- *acs (with the primers ACS-f / ACS_r) encoding acetyl-CoA synthase for the reductive acetyl CoA pathway*

The unsuccessful PCR amplifications are in agreement with the metagenomic data that did not highlighted any other CO₂ fixation pathway in addition to the CBB cycle (now more clearly indicated lines 207, 256-257). Indeed, we only detected genes coding for the phosphoribulokinase (prkB) and subunits of ribulose-1,5-bisphosphate carboxylase (rbcl and rbcS, respectively).

We now indicated in the “Methods” section that cbbL and cbbM were not the only genes that have been considered (lines 606-612). All the functional genes related to inorganic carbon assimilation that were tested by using selective PCR amplifications are now individually mentioned in the caption of Supplementary Table S4.

Line 278: What were the potential roles for the two detected archaeal populations?

As described in the caption of Supplementary Fig. S13, the two single OTUs belonging to the Thaumarchaeota and the Euryarchaeota phyla form clades with other environmental sequences but are very distantly related to sequences of cultivated species (respectively 91 and 92% identities at the level of their partial 16S-rRNA gene sequences). Consequently it is difficult to infer a putative role of the archaeal populations from the physiology of the closely related described taxa. However, because the quantity of amoA gene was of the same order of magnitude than the thaumarchaeotal 16S-rRNA gene during the whole microbiological survey (new Supplementary Fig. S14), we can assume that the Thaumarchaeota species likely corresponds to ammonia-oxidizing archaea (AOA), diagnosed using amoA. As the AOA accounted for ~20% of the archaea, the major archaeal fraction belongs accordingly to Euryarchaeota with hence prominence of the TMEG lineage in the archaeal diversity. Although numerous environmental species close to the TMEG lineage possess the mcr gene, a functional marker for methanogenesis, this gene was not detected in HN-04 groundwater, suggesting that this species is not methanogenic. Nonetheless environmental sequences close to the clade of Euryarchaeota-related OTUs were retrieved from oil sand formation water, subseafloor sediments in methane seeps or with gas hydrate potential, and petroleum reservoir (accession numbers GU553457, JF789592, JQ989763, JQ989723, FN429784JN123612; Supplementary Fig. S13), hence suggesting, as for bacteria, a key role of organic

compound in supporting the aquifer community.

The new Fig. S14 was added to the Supplementary Material to support this statement. We also completed the Material and Methods Section and Table S9 along with the Supplementary reference list with information on qPCR experiments performed on archaea.

Line 289: This is a similar statement as made in line 148.

We agree and modified the sentence accordingly (lines 319-322): “The gas injection altered geochemical conditions by introducing acidified waters, which prompted rock dissolution resulting in ion release¹⁴ (Table S3). The release of ions may have been beneficial for certain bacterial types as suggested by Mantel correlations (Fig. 5; Table S5).”

Line 292: Ferrous iron likely constituted an electron donor for the populations. This has not been proven.

*We added “likely” to the former sentence (line 322). To support the likely use of ferrous iron, we looked in the metagenomic data for the presence of the *cyc2* gene, which is present in Sideroxydans and for which there is evidence of its involvement in Fe-oxidation. The high number of hits (now provided in Supplementary Table S8) unambiguously confirmed the overrepresentation of iron oxidizers in March 2012 in the aquifer, in agreement with the bloom of species affiliated to *S. lithotrophicus*. In addition, we now provide in the Supplementary Fig. S2, the temporal evolutions of Fe^{2+} and $Fe(\text{total})$ showing the consumption of Fe, mainly present as Fe^{2+} , following its release from the rock.*

Line 295: Why was zinc and magnesium release only beneficial for heterotrophs?

*Zn and Mg were not only beneficial for heterotrophs and autotrophs are also considered in the discussion. Our aim was to establish a link between the main metals released from the basalt and the putative metabolisms that were highlighted by metagenomic analysis. Main metals included Fe, Zn, Mg, Mn (and Ca) and as described lines 326-327, they are all needed as cofactor in many of the enzymes highlighted by metagenomic analysis that relate to both heterotrophs and autotrophs (Table S7). Most markers of the degradation of aromatic compounds are dependent on Zn (DLH, *tas*), Fe (*ligB*, *gloA*, *nirD*, *hcaE*), Mn (*hgdB*) and Mg (*hgdB*), however, Mg was also shown to be beneficial for autotroph. Mg^{2+} was indeed stated to be mandatory for the activation of the RuBisCO encoded by the *cbbL* and *cbbM* genes (Figs. S7-S8) that is crucial for carbon fixation (lines 333-334). We rewrote the section to make it clearer.*

Line 307: Basalt dissolution was not involved in the general bloom of bacteria. Instead, I would say that a bloom may have occurred as a consequence of the ions released by the dissolution.

We agree and we changed the sentence accordingly (lines 339-340): “Mantel test and canonical correlations confirmed ions released by basalt dissolution stimulated the growth of autotrophic and heterotrophic bacteria in the aquifer (Fig. 5, Table S5).”

Line 310: How can CO₂ fixation be linked to heterotrophy? If the Desulfotomaculum genus is using the reductive carboxylate cycle then it is an autotroph. Also, please see line 322 where it is stated “heterotrophic rates of CO₂ fixation”.

*As indicated in Fig. 6, Supplementary Table S7 and in text (lines 281-283), the key markers retrieved by metagenomic analysis point to the oxidative Acetyl-CoA pathway instead of the reductive one. This pathway involve the bifunctional CO dehydrogenase/acetyl-CoA synthase (*Cdh*) subunits for which an increase of the signal intensity was detected during the bloom of Firmicutes in May 2012. This pathway is characteristic of strict anaerobes degrading aromatic rings. The oxidation of Acetyl-CoA is in that case used to fuel the activation steps of ring destabilization as the degradation of aromatic compound is an ATP-dependent reaction (Bozinovski et al., 2014). We hence postulate that the Desulfotomaculum species*

blooming at that time period is likely an heterotroph. Notably, Desulfotomaculum profundum, its closest cultivable relative, was isolated from a community capable of degrading ethylbenzene, toluene and benzene (Berlendis et al., 2016).

Concerning the link between CO₂ and heterotrophy, it has been recently shown that strictly anaerobic sulfate reducers and other Peptococcaceae typically lack enzymes of the glyoxylate cycle to build up their biomass (Kosaka et al., 2006; Chivian et al., 2008). As a consequence, they assimilate acetyl-CoA through reductive carboxylation to pyruvate by the pyruvate: ferredoxin oxidoreductase (Heider and Fuchs, 1997), hence consuming CO₂ which is reduced in downstream anabolism. Accordingly, several studies have shown that such an heterotrophic CO₂ assimilation can be performed at high rates through this process (Winderl et al., 2010; Taubert et al., 2012). Additionally, as observed in a wide variety of habitats, from oil reservoirs and coal deposits to contaminated groundwater and deep sediments, CO₂ can be used as electron acceptor to sustain intermediate steps in the microbial transformation of hydrocarbons (Jiménez et al., 2016), hence also linking CO₂ consumption with heterotrophy.

We reworked the corresponding paragraphs (lines 343-347) to make this point clearer. A strain of Desulfotomaculum profundum was isolated from the Hellisheidi groundwater sampled during the present survey. Its genome was sequenced and is currently examined to further document its carbon assimilation pathways.

Cited references:

- Berlendis, S. et al. (2016) *Desulfotomaculum aquiferis* sp. nov. and *Desulfotomaculum profundum* sp. nov. isolated from a deep natural gas storage aquifer. *Int. J. Syst. Evol. Microbiol.*, doi: 10.1099/ijsem.0.001352.
- Bozinovski, D. et al. (2014) Metaproteogenomic analysis of a sulfate-reducing enrichment culture reveals genomic organization of key enzymes in the m-xylene degradation pathway and metabolic activity of proteobacteria. *Syst. Appl. Microbiol.* 37, 488-501.
- Chivian, D. et al. (2008). Environmental genomics reveals a single-species ecosystem deep within earth. *Science* 322, 275-278.
- Heider, J. & Fuchs, G. (1997). Anaerobic metabolism of aromatic compounds. *Eur. J. Biochem.* 243, 577-596.
- Jiménez, N., Richnow, H.H., Vogt, C., Treude, T. & Krüger, M. (2016) Methanogenic hydrocarbon degradation: evidence from field and laboratory studies. *J. Mol. Microbiol. Biotechnol.* 26 (1-3), 227-42.
- Kosaka, T. et al. (2006). Reconstruction and regulation of the central catabolic pathway in the thermophilic propionate-oxidizing syntroph *Pelotomaculum thermopropionicum*. *J. Bacteriol.* 188: 202–210.
- Taubert, M. et al. (2012) Protein-SIP enables time-resolved analysis of the carbon flux in a sulfate-reducing, benzene-degrading microbial consortium. *ISME J.* 6, 2291-2301.
- Winderl C., Penning H., von Netzer F., Meckenstock R.U. & Lueders T. (2010) DNA-SIP identifies sulfate-reducing Clostridia as important toluene degraders in tar-oil-contaminated aquifer sediment. *ISME J.* 4, 1314-1325.

Line 390: Please define “at great depth”.

Depth values for the pumps and the casing in HN-01 and HN-04 are now provided (lines 453-459).

Line 391: What do the authors mean by “reservoir conditions”?

We agree that the use of the words “reservoir conditions” was not appropriate as the groundwater sampling was not performed under in situ conditions for the microbiological survey. We used instead “allow groundwater sampling in the targeted basaltic formation” (lines 459-460).

Line 408: Is there any evidence that loss of groundwater pressure would result in precipitation of dissolved ions, altering both pH and redox potential?

We acknowledge that a crucial issue of monitoring is the quality of the sampling and that loss of pressure can be critical as pressure impacts mineral solubility. Fluids samples from deep wells and subsequent sample treatment prior to gas and liquid analysis require special equipment and sampling techniques to

account for the relatively high temperatures, pressures, and potential gas content present at depth. With this aim, a piston-type downhole bailer was designed, constructed and tested in the framework of the Carbfix project (Alfredsson et al., 2011; Wolff-Boenisch et al., 2014). It allowed comparing sampling performed under reservoir conditions with the routine ones. Comparison of the analytical results obtained by both approaches did not show discrepancies, thus allowing to infer that no major changes in groundwater chemistry occurred during depressurization, at least for the parameters that were tested (pH, redox, alkalinity). Geochemical modelling was also used to investigate the probability of decarbonation and concomitant carbonate scaling during sampling.

Cited references:

- H.A. Alfredsson, D. Wolff-Boenisch, A. Stefánsson (2011) CO₂ sequestration in basaltic rocks in Iceland: Development of a piston-type downhole sampler for CO₂ rich fluids and tracers. *Energy Procedia* 4, 3510-3517.
D. Wolff-Boenisch, K. Evans (2014) Review of available fluid sampling tools and sample recovery techniques for groundwater and unconventional geothermal research as well as carbon storage in deep sedimentary aquifers. *Journal of Hydrology* 513, 68-80.

Line 410: I do not think “confrontation” is the correct word. Please edit.

We are now using the word “comparison” (line 479).

Line 433: How were the obtained cycle thresholds (Ct) defined as not significantly different to those obtained when quantifying the plasmid alone? Please give details.

The Ct values obtained during the inhibition tests allowed us to assess if the extracted DNA inhibited the qPCRs. For this reason comparison of the Ct values of samples containing plasmid + extracted DNA were compared to Ct values of samples containing plasmid alone. The percent inhibition (or actually % efficiency) can then be calculated according to the following formula:

$$1 - ((C_{t_{\text{sample}}} - C_{t_{\text{standard}}}) / C_{t_{\text{standard}}}) \times 100$$

Efficiencies lower than 95% were indicative of inhibited samples that were then diluted or removed from the essay. Details are now given (lines 504-508).

Lines 450 and 492: Give the DNA concentrations (and not volume).

DNA concentrations are now also provided (lines 521-522 and 562-563).

Line 576: State what “Electrospray ionization Fourier transform ion cyclotron resonance/mass spectrometry” was used to detect.

We added a sentence to state the reason for the use of FT-ICR- MS (lines 659-661): “In order to assess the molecular diversity of organic compounds present in groundwater through time, non-targeted ultrahigh-resolution molecular analysis of the solvent-accessible organic fraction were performed.”

Line 656: The “2” should be in subscript.

Line 666: The reference title should not be capitalized.

This is now fixed (line 758 and 768-770).

Line 782: Please fix “This is IPGP contribution n° XXX”.

The publication number is delivered by our institute once a paper is accepted for publication. Accordingly, this can only be fixed later on.

Fig. 3: Are the size of the circles representative of up to the given percentage i.e. the smallest circle represents 1 to 5% while the second smallest is 5 to 10%? Please give more details.

The size of the circles used in Fig. 3 represents the exact percentage of the OTU abundance. In the size scale, some reference points are given for values ranging from 1 to 40% and encompassing encountered OTU abundance values. We modified the caption of Fig. 3 to make this point clearer.

Fig. 5: Are these not genes and therefore, should be in italics?

Gene names are now in italics in Fig. 6 (and associated caption) and in Table S7.

Reviewers' Comments:

Reviewer #1:

Remarks to the Author:

This MS is much improved from the previous version, highlights include a more straightforward presentation of results, and an interesting discussion on the possible sources of aromatic compounds that stimulated the community during its 'bloom' phase.

There are still some sections in the results that could use clarification.

Specifically:

l156, provide the range of OTUs here from what to approx 100

l161. Why is Zn cited as an example? There are no known microbial redox reactions that depend on Zn. What Fe or Mn or reduced S species?

l176, this is an overstatement, there is no direct evidence mtoA is responsible for Fe-oxidation in S.l., it is all based on inference. Better to say 'putatively linked to Fe-oxidation'

;182-183, sentence is confusing, was the most abundant motif, or the most abundant motif related to PhnJ. This is important, since it's well known P binds strongly to Fe-oxides, so once biogenic oxides were being formed P would become limiting, thus increased expression of P-acquiring transporters is consistent.

l201-202, rewrite sentence for clarity.

l217-221, rewrite sentence for clarity

l259, favored when in May or in March?

l260-262, unclear what is meant by this sentence.

Reviewer #2:

Remarks to the Author:

The language is much improved but there are still small points that need addressing. I have listed some below but once again, have not continued through the whole document (I read the introduction and then started on the methods).

Line 44: Please edit this sentence for clarity.

Line 62: Please edit this sentence for clarity (the last part from 'microbial populations').

Line 81: 'Identify' and not 'identity'.

Line 81: '...compared to the native microbial...'

Line 438: 'Groundwater sampling was carried...'

Line 439: '...six groundwater samples for two...'

Line 442: '...already begun for a few weeks...'

Line 449: `...serve as a control well.'

Line 450: `...gas was mixed...'

Line 454: '... allowed an investigation of the extent that the HN-01..."

Line 458: `...superficial aquifer, the CarbFix1 injection site wells are cased...'

The authors have addressed the majority of my concerns regarding statistical analyses. However, Pearson correlations of chemical data and relative abundance of individual annotated phyla could be carried out to provide support that e.g. "The decrease in the Gallionellaceae number of representatives in May'12 is likely related to pH, Fe²⁺ or oxygen concentrations no longer compatible with their development" (lines 325-326).

The authors acknowledge that the planktonic populations (as compared to biofilms) may represent a biased view of the microbial diversity. I still think that a comment in the discussion about the Luef et al work on ultra-small cells suggests the use of a 0.22 µm filter for cell capture may also represent an underestimation of the diversity would be good.

Revised manuscript NCOMMS-16-25435A

“High reactivity of deep biota under anthropogenic CO₂ injection into basalt”

Responses to the Referees

We would like to thank once again the Referees for careful reading of the manuscript and their constructive comments. Changes and additions were made accordingly in this second revised version. Responses to the Referees' comments are provided below in blue and in italics. The line numbers refer to the revised manuscript.

Reviewer #1 (Remarks to the Author):

This MS is much improved from the previous version, highlights include a more straightforward presentation of results, and an interesting discussion on the possible sources of aromatic compounds that stimulated the community during its 'bloom' phase.

There are still some sections in the results that could use clarification.

Specifically:

l156, provide the range of OTUs here from what to approx 100

The range of OTU variations is now specified lines 156-157.

l161. Why is Zn cited as an example? There are no known microbial redox reactions that depend on Zn. What Fe or Mn or reduced S species?

Among all the metals thought to influence microbial blooms, both as enzymatic cofactor or source of energy, Zn was cited as one of the most important increase in concentrations. We removed this portion of the sentence to avoid misunderstanding.

l176, this is an overstatement, there is no direct evidence mtoA is responsible for Fe-oxidation in S.I., it is all based on inference. Better to say 'putatively linked to Fe-oxidation'

“Putatively” was added to the sentence line 181.

;182-183, sentence is confusing, was the most abundant motif, or the most abundant motif related to PhnJ. This is important, since it's well known P binds strongly to Fe-oxides, so once biogenic oxides were being formed P would become limiting, thus increased expression of P-acquiring transporters is consistent.

We thank Reviewer#1 for his/her comments. We simplified the sentence by changing “related to” to “was” and we amended the paragraph with this interesting suggestion (lines 187 to 190).

l201-202, rewrite sentence for clarity.

Sentence was rewritten (lines 209 to 210).

l217-221, rewrite sentence for clarity

Sentence was rewritten (lines to 226 to 235). For sake of clarity and consistency, we now document in the

captions of Supplementary Figs. 7 to 9, the order and family of each species present in the phylogenetic trees.

l259, favored when in May or in March?

“In May’12” was added to the sentence (line 269).

l260-262, unclear what is meant by this sentence.

Sentence was clarified (lines 272 to 274).

Reviewer #2 (Remarks to the Author):

The language is much improved but there are still small points that need addressing. I have listed some below but once again, have not continued through the whole document (I read the introduction and then started on the methods).

Line 44: Please edit this sentence for clarity.

The sentence was edited (lines 44 to 47).

Line 62: Please edit this sentence for clarity (the last part from ‘microbial populations’).

The end of the sentence was edited (lines 63 to 64).

Line 81: ‘Identify’ and not ‘identity’.

We did not make any change because we are talking about genes of function and identity.

Line 81: ‘...compared to the native microbial...’

The change was made (line 81).

Line 438: ‘Groundwater sampling was carried...’

The change was made (line 453).

Line 439: ‘...six groundwater samples for two...’

The change was made (line 454-455).

Line 442: ‘...already begun for a few weeks...’

The change was made (line 458).

Line 449: ‘...serve as a control well.’

The change was made (line 464).

Line 450: ‘...gas was mixed...’

The change was made (line 465).

Line 454: ‘... allowed an investigation of the extent that the HN-01...’

The change was made (line 468 to 470).

Line 458: ‘...superficial aquifer, the CarbFix1 injection site wells are cased...’

The change was made (line 473).

The authors have addressed the majority of my concerns regarding statistical analyses. However, Pearson correlations of chemical data and relative abundance of individual annotated phyla could be carried out to provide support that e.g. “The decrease in the Gallionellaceae number of representatives in May’12 is likely related to pH, Fe²⁺ or oxygen concentrations no longer compatible with their development” (lines 325-326).

Pearson correlation coefficients between chemical data and relative abundance of individual annotated phyla were estimated and values are now provided line 338 to 341 but also lines 162-164 and 175-178. The “Material and methods” section was modified accordingly (line 618 to 620).

The authors acknowledge that the planktonic populations (as compared to biofilms) may represent a biased view of the microbial diversity. I still think that a comment in the discussion about the Luef et al work on ultra-small cells suggests the use of a 0.22 µm filter for cell capture may also represent an underestimation of the diversity would be good.

A comment was added in the discussion to mention the bias induced by groundwater filtration using 0.22 µm filters (lines 367 to 369).

Reviewers' Comments:

Reviewer #1:

Remarks to the Author:

The authors have met all my comments. Thanks.

Reviewer #2:

Remarks to the Author:

The authors have dealt with the previous points and I have no further comments.